# Online college English education in Wuhan against the COVID-19 pandemic: Student and teacher readiness, challenges and implications

**Cuiying Zou**, **Ping Li, Li Jin** *

School of Foreign Languages, Wuhan Business University, Wuhan, People's Republic of China

* janet202104@126.com

**Data Availability Statement:** All relevant data are within the manuscript and its Supporting Information files.

## Abstract

Online education, including college English education, has been developing rapidly in the recent decade in China. Such aspects as e-readiness, benefits and challenges of online education were well-researched under normal situations, but fully online language teaching on a large-scale in emergencies may tell a different story. A survey of 2310 non-English-major college students and 149 English teachers from three types of twelve higher education institutions in Wuhan was conducted to evaluate their readiness for online English education during the COVID-19 pandemic, to figure out challenges encountered by them and to draw implications for future online college English education. Quantitative statistics gathered using two readiness scales adapted from previous studies showed that both cohorts were slightly below the ready level for the unexpected online transition of college English education. The overall level of readiness for students was 3.68 out of a score of 5, and that for teachers was 3.70. Individual differences were explored and reported. An analysis of qualitative results summarized six categories of challenges encountered by the students, i.e. technical challenges, challenges concerning learning process, learning environment, self-control, efficiency and effectiveness, and health concern. Though the students reported the highest level of readiness in technology access, they were most troubled by technical problems during online study. For teachers, among three types of challenges, they were most frustrated by pedagogical ones, especially students' disengagement in online class. The survey brought insights for online college English education development. Institutions should take the initiative and continue promoting the development of online college English education, because a majority of the respondents reported their willingness and intention to continue learning/teaching English in online or blended courses in the post-pandemic period. They are supposed to remove technical barriers for teachers and students, and assess the readiness levels of both cohorts before launching English courses online. Institutions should also arrange proper training for instructors involved, especially about pedagogical issues. Language teachers are suggested to pay special attention to students' engagement and communication in online courses.

**Funding:** This work was funded by Education Research Project of Department of Education of Hubei Province (Grant No. 2020688), Education Research Project of Wuhan Business University (Grant No. 2021N031) and Scientific Research Project of Wuhan Business University (Grant No. 2021KY007). The funders had no role in study design, data collection and analysis, decision to publish, or preparation of the manuscript.

**Competing interests:** The authors have declared that no competing interests exist.

# Introduction

The COVID-19 pandemic has caused enormous transformations in various aspects of the society since its outbreak. One of the greatest impacts is its disruption in education. In China, the spring semester in 2020 for schools at all levels was first postponed [1] and then moved online [2], which was considered a quick-fix to some extent. However, the recent decade has witnessed enormous advancements in online education in China, especially at the tertiary level, which has laid a solid foundation for the online transition. According to statistics, in response to the Ministry of Education's (MOE) instructions on the deployment of Higher Educational Institutions (HEIs) online teaching to enable students to resume their studies remotely, by February 2nd there had been 22 online platforms in China providing 24,000 online HEI courses free of charge, covering 12 disciplines at undergraduate level and 18 disciplines at higher vocational education level. Teachers had been encouraged to teach online by using various online resources including Massive Online Open Course (MOOC) platforms [2]. With everything moved online, *College English*, a compulsory course for first- and second-grade non-English majors from HEIs offering degree programs, was no exception. This pandemic-prompted nation-wide online college English education was unprecedented in China. English teachers were forced to make what was thought to be impossible possible at short notice (mostly two weeks' preparation). This online teaching in the face of COVID-19, or online triage [3], was carried out without sufficient need analysis or readiness evaluation from both the learning and the teaching sides. It was different from planned and well-prepared online education.

Online education offers numerous advantages over traditional face-to-face (F2F) education including flexibility, accessibility, independence, interactivity, multimodality, cost-effectiveness, ubiquitous learning, convenience, and learner-centeredness [4], It has been gaining increasing popularity in recent decades. Even before COVID-19, there was already high growth and adoption in education technology, with global edtech investments reaching US $18.66 billion in 2019 and the overall market for online education projected to reach $350 billion by 2025 [5]. Then with sudden closures of schools, colleges and universities to ensure the safety of communities in many parts of the world, shifting online became the only possible solution to make education continue. Various technological tools experienced a soar in usage since the breakout of the pandemic.

As for language learning environment, however, online instruction has just begun to enjoy the same popularity already experienced within other disciplines in the latest decade. The term *online language learning* (OLL) can refer to a number of learning arrangements: a Web-facilitated class, a blended or hybrid course, a fully virtual or online course [6]. These delivery formats, which can be synchronous, asynchronous or both, utilize a variety of technologies, including (desktop) videoconferencing, computer-mediated communication (CMC) tools, and Web 2.0 technologies [6–8]. Online language education has affordances that are different from F2F courses. First of all, it is flexible, adaptive and allows for enhanced, individualized, and authentic materials. Secondly, it can take advantage of communicative tasks and multilingual communities. Lastly, it can also foster and take advantage of autonomous learning and learner corpora [9]. Some studies have proved the effectiveness of online language learning [10–12], and some explored such themes as online language learning assessment [13–15], learners' perspectives [16, 17], and language teachers' professional development [18–20]. More recent delivery modes of online language education include formal online language courses, virtual worlds, Language Massive Open Online Courses (LMOOCs), online language learning communities and mobile apps for language learning [21]. In China, LMOOCs, especially for English, though at its infancy, has been gaining popularity in recent years. By searching for the key word "英语" (English) on the top five MOOC platforms in China, a total of 1391 courses

appeared, including general English such as *College English*, *English Writing* and English for more specific purposes. Many HEIs are developing MOOCs and SPOCs for English education online, but few of them implement fully online teaching.

Generally speaking, for college English courses in Wuhan, online learning is currently acting as a complementary means to classroom teaching. Learning platforms coming along with textbooks and self-developed MOOCs or Small Private Online Courses (SPOCs) are the mainstream tools for English teachers to implement online or blended teaching [22, 23]. Problems may occur, but usually at low frequency and do not cause too many anxieties because there is classroom teaching. However, during the pandemic, online English learning was the only compulsory means rather than a complementary one. Problems occurred frequently, especially at the beginning, and caused anxieties among students and teachers. Some problems might be specific to the pandemic context, others might be common even in the non-pandemic period. Therefore, now that the semester has terminated smoothly and successfully, lessons can be drawn for future development of online college English education. This research aims to draw implications for the development of online college English education through measuring readiness levels of students and teachers for the online transition and probing into the problems they met in this particular context. The following research questions were explored:

1. To what extent were college students and teachers ready for online English education during the pandemic semester? Are there any individual differences?

2. What challenges did teachers and students encounter during the pandemic semester?

3. What were teachers' and students' perceptions towards future online English education?

## Literature review

### Technology acceptance model

Technology acceptance model (TAM) was put forward by Davis [24] and the construct was defined as a set of technology-related attitudes and beliefs that explain a person's intentions to use and actual use of technology. The model was designed to explain technology acceptance across a broad range of information technologies, and it suggests that a number of factors influence the users' attitudes towards technology and decisions about how and when to use a new technology, notably perceived usefulness (PU) and perceived ease of use (PEOU). According to Davis [24], PU is the degree to which a person believes that using a particular system would enhance his or her performance and PEOU is the degree to which a person believes that using a particular system would be free from effort.

The TAM model has been frequently empirically tested and proved to be a useful theoretical model to understand and explain use behaviour in information system in many contexts and fields [25]. One of these contexts is education [25–28]. TAM was found to be the most common theoretical framework in online learning acceptance [29]. Closely related to the concept of technology acceptance is the readiness to use technology in learning or teaching, which can be called online learning/ teaching readiness and will be discussed in the following sections.

### Student readiness for online English learning

Warner, Christie, & Choy [30] have defined readiness for online learning or e-readiness as a measure of students' inclination toward online delivery modes versus F2F instruction, their competence and tendency to utilize electronic communication, and their ability to undertake

self-directed learning. Several studies concluded the significance of e-readiness in online learning from different perspectives. Moftakhari [31] claimed the success of online learning entirely relied on learners' and teachers' readiness levels, which seems to be absolute. Piskurich [32] believed low readiness level was the main reason for failure in online learning. Students' e-learning readiness was proved statistically as a significant predictor of their satisfaction for online instruction [33–35]. Therefore, assessing student readiness for online learning is highly relevant prior to delivering a course online either fully or hybridly and promoting student readiness is essential for successful online learning experiences [36]. A typical readiness assessment will assess the student's ability in adapting to technological challenges, collaborative learning and training as well as the synchronous and asynchronous self-paced learning [37]. A number of studies developed and validated online learning readiness scales, covering similar but not the same dimensions [38–41]. Computer self-efficacy, one main component of online learning readiness, is defined as individuals' perceptions of using a given technology and individuals' ability to use the technology [40]. Knowles [42] defined self-directed learning, another component, as a process in which individuals take the initiative, with or without the help of others, in diagnosing their learning needs, formulating learning goals, identifying human and material resources for learning, choosing and implementing appropriate learning strategies, and evaluating learning outcomes. Learner control, which is also an important factor in online learning readiness, is defined as the degree to which a learner can direct his or her own learning experience and process [43]. Motivation can be divided into two categories: intrinsic motivation, which refers to doing something because it is inherently interesting or enjoyable, and extrinsic motivation, which refers to doing something because it leads to a separable outcome [44]. Motivation towards online learning means having intrinsic and extrinsic desire in using online learning. Another essential dimension for overcoming the limitations of online communication is online communication self-efficacy [40], being defined as how well the learners can express their feelings and measuring the level of understanding of the communication language and culture [45].

Online language learning is different from online learning of other subjects. Unlike other subjects, language is both the medium of instruction and the subject matter of online learning. In the learning process, the learners are supposed to listen, speak, read and write in the language they are learning. Therefore, whether the online learning environment can provide opportunities for the learners to use the language and whether the learners feel free to use it online determines the success of the language course. Tylor and David [46] claimed that little attention had been paid to learner preparedness for online language learning and developed a self-assessment survey tool with indicators of learner autonomy, computer self-efficacy, attitude towards online learning, motivation, and English language self-efficacy. It shed some lights for researchers in this area, but further validation of their model is needed. Marzieh and Salman [47] identified the factors affecting e-learning acceptance and readiness in the context of foreign language learning. Their findings indicated the complex relationships between the perceived usefulness, perceived ease of use, e-learning motivation, online communication self-efficacy and language learners' acceptance and readiness of e-learning. One limitation of their study is the relatively small sample size. Mehran et al [48] reviewed the limited number of studies on readiness for online language learning and summarized a set of influencing factors which can be broken down into two general categories: demographic variables which incorporate gender, age, grade, nationality, field of study, and technological accessibility/ownership versus non-demographic variables which encompass learner autonomy, motivation, learning style, attitude toward e-learning, language self-efficacy, technological acumen, and online communication skills. Based on these studies, they carried out a survey and found out the students in Osaka University unwilling to take online courses, either fully online or blended.

Further studies are still needed, because nationality is a variable [46]. To the best of the researchers' knowledge, there has been no study conducted to assess Chinese college students' readiness for online English learning. Fully online teaching for college English was implemented because of the pandemic, and the researchers took this opportunity to carry out the study, in order to have a rough idea about college students' readiness for online English learning.

## Teacher readiness for online language teaching

Concerning instructors, teaching online requires a reconstruction of their roles, responsibilities, and practices [36]. Teacher readiness, or faculty readiness refers to the willingness to prepare, effectively design and facilitate courses within an online environment [49]. Adnan's [34] study proved a significant relationship between faculty readiness and satisfaction. Understanding the level of teacher readiness for online teaching in an institution is a key component in the journey to successfully facilitating online courses and programs [49]. The level of online instructors' e-readiness was evaluated using three scales: technical readiness, lifestyle readiness and pedagogical readiness to the e-learning system environment and it was found that the cohort surveyed were more technologically ready than in lifestyle and pedagogically [50]. These studies all focused on online teaching in general. However, specific to language teaching, online teachers need different skills than those who are trained to teach languages in a F2F classroom and they also require different skills compared to online teachers of other subjects [18]. Hampel and Stickler's research was one of the earliest comprehensive studies focusing on the pedagogical aspects of online language teaching. They specified seven levels of skills required for online language teaching in a pyramid style from lower-level skills such as basic ICT competence, specific technical competence for the software, dealing with constraints and possibilities of the medium to higher-level skills such as online socialization, facilitating communicative competence, creativity and choice and own style [18]. Compton [19] reviewed Hampel and Stickler's skill pyramid critically and proposed an alternate framework for online language teaching skills which consists of three dimensions: technology in online language teaching, pedagogy of online language teaching and evaluation of online language teaching. These studies are mainly theoretical, focusing on explaining why certain skills are important, without actually measuring whether the language teachers are ready for online language teaching. Based on the findings in previous studies, this research is going to measure the readiness level of English teachers in Wuhan during the pandemic.

## Online language teaching during the COVID-19 pandemic

The COVID-19 pandemic brought about studies on emergency online language teaching. Gacs et al. [3] and Ross and DiSalvo [51]) studied how higher education language programs had swiftly moved from traditional to online teaching. Moorhouse [52] and Moser et al. [53] investigated how the form of language instruction has changed during the COVID-19 pandemic. Chung and Choi [54] examined a case of an English language program in South Korea to investigate how the sudden transition to online language teaching has influenced language instructors' teaching and assessment practice. Maican and Cocoradă [55] analyzed university students' behaviors, emotions and perceptions associated with online foreign language learning during the pandemic and their correlates. There were also a few studies with the context of China. Lian et al [56] surveyed 529 Chinese university students on their perceptions of authentic language learning (AULL), self-directed learning (SDL), collaborative learning (CL), and their English self-efficacy (ESE) during the online learning period of the COVID-19 pandemic. Zou et al [57] explored three English writing teachers' engagement with online formative

assessment during COVID-19 in three universities in China, and identified three types of teacher engagement, namely disturbing, auxiliary, and integral engagements. Gao and Zhang [58] analyzed in-depth interviews with three English teachers from a Chinese university and found that teachers had clear cognitions about features, advantages, and constraints of online English teaching.

Studies on language learning during the pandemic keep emerging, but no one has studied students' and teachers' readiness for the transition from traditional teaching to fully online teaching or the actual problems they met in the process. Therefore, based on the review above, this research, with a combination of the TAM, the online learning components and online language teaching skills introduced by previous research as the theoretical framework, carried out a survey among college non-English-major students and college English teachers in Wuhan. The aim was to assess their readiness levels for shifting from classroom teaching to fully online teaching during the pandemic semester, to analyze the actual challenges they encountered, to understand their perceptions towards future online English learning and teaching, and to draw implications for future development of online college English education. It is hoped this research can bring some insights for the field of online language education as well as remote learning and teaching in emergency situations.

## Methodology

### Ethical statement

This research was investigation-oriented without revealing any specific personal information, so no ethical agreement was needed. Participants were recruited on a voluntary basis, and there was a sentence-"The survey is anonymous, but if you finish it, the researchers will understand it as a formal consent for using your responses in future publications of the research."-in the introduction part of the survey. Therefore, no extra formal consent was obtained from the participants. Upon completion of the questionnaires or interviews, they automatically granted use of their responses.

### Instrument design

Data in this research were collected using two questionnaires, one for students and another for teachers. Both consist of a demographic information form, a readiness scale and two open-ended questions. To avoid possible misunderstanding, both questionnaires were presented in Chinese. Translation of the questionnaires were done by one of the researchers who has a master's degree in translation, and double checked by a colleague also with a master's degree in translation.

The student readiness scale (SRS) was an adapted version of the Online Learning Readiness Scale (OLRS) developed and validated by Hung et al. in 2010 [40]. Both the English and Chinese versions of the scale were shared by the original authors and permission was given to adapt the scale. The OLRS evaluates online learner readiness from five dimensions: computer/internet self-efficacy, self-directed learning, learner control, motivation for learning and online communication self-efficacy. However, stakeholders' technology access, and infrastructure may greatly impact what is possible [3], especially during the emergency transition when all students and teachers stayed at home without access to any of the school facilities. Therefore, a sixth dimension of technology access with three items was added from a scale developed by Watkins et al. [39]. The wording of the items from the original scales were changed to refer more specifically to English learning. For example, "I carry out my own study plan" was changed into "I carry out my own English study plan". All items used a five-point Likert-type scale, ranging from "1 = strongly-disagree" to "5 = strongly-agree".

The teacher readiness scale (TRS) used the three dimensions included in Gay's [50] online instructor e-readiness scale, i.e. technical readiness, lifestyle readiness and pedagogical readiness. 11 items were kept from Gay's scale with modifications made in diction to refer specifically to English teaching. Another 5 items were added to include the skills required for online language teachers discussed in previous studies [18, 19]. All items used a five-point Likert-type scale, ranging from "1 = strongly-disagree" to "5 = strongly-agree".

The two questionnaires were initially piloted to check clarity of the language used and to ensure the reliability of the two scales in the local context. Both scales were statistically reliable, with Cronbach's Alpha coefficients being 0.961 and 0.859 respectively. Improvements were made in light of the comments from pilot respondents and two colleagues with expertise in questionnaire design. Improvements included removing one item which was overlapping with another one from the TRS and clarifying language ambiguities. For instance, one item "当我的电脑软硬件出现技术问题时, 有人和/或资源给我提供帮助。" was modified as "在线教学过程中出现技术问题时, 有人(同事, 家人, 平台技术支持)和/或资源(手册, 视频)给我提供帮助。". The finalized version of the SRS had 22 items in six dimensions and that of the TRS 15 in three. As alpha coefficients of 0.7 or over are considered acceptable [59], both scales were proved to be reliable and valid with overall Cronbach's Alpha coefficients being 0.974 and 0.957 respectively with the main samples, and each dimension had an Alpha coefficient greater than 0.8. The open-ended questions corresponded each other in the two questionnaires to gather equivalent information from both cohorts. The questions were: "What challenges have you encountered in your online English learning/teaching during this semester?" and "Are you willing to learn/teach English either in a fully online or blended course in the future? Why?".

In addition to the two questionnaires, a semi-structured interview was designed to get an overall view of the situation before and during the pandemic semester for the sake of better understanding of the descriptive analysis of the survey. There were five questions in the interview, focusing on five aspects, i. e. the situation of online English teaching before the pandemic, the teaching modes and online teaching platforms used during the pandemic, assessment criteria for the online English course and possible plans to develop fully online or blended college English courses. The interview was conducted in Chinese orally or in written form.

## Data acquisition

Participants were recruited using purposive sampling and convenience sampling. There are a total number of 46 HEIs in Wuhan offering degree programs, among which 8 are directly under MOE, 15 under Hubei Provincial Department of Education and 23 non-governmental. All these HEIs offer college English courses to first- and second-year students. In order to involve universities at all levels, the director of the School of Foreign Languages from the researchers' university sent invitations to her counterparts from 19 institutions (four directly under MOE, ten under Hubei Provincial Department of Education and five non-governmental) purposively to recruit teacher and student respondents who teach or learn college English courses. All of them accepted the invitation to help without promising a satisfactory result because participation was voluntary and responses were anonymous.

The questionnaires were disseminated online from July 24[th] to August 2[nd], 2020 after the semester terminated in all universities. Both were posted on one of the most widely-used online survey platforms in China powered by *https://www.wjx.cn/* and only those invited by their college English teachers got the access to participate. The survey was set to allow only one submission for the sake of data integrity.

The interviews were conducted during the same time period by the researchers with the personnel in charge of college English education from the sampled universities or colleges on a one-to-one basis through email or online chatting tools. Written answers were copied directly. Oral ones were transcribed automatically by chatting tools first and then checked by one of the researchers before further analysis.

Throughout the survey, personnel in charge of college English education from 16 universities participated in the interview. Among these universities, four were excluded because researchers received no teacher response or less than ten student responses. Among 2351 students who completed the student questionnaire, 15 were from universities from which no interview was conducted, and 26 completed the questionnaire in less than 60 seconds (the researchers tried to finish the questionnaire as fast as they could and determined the minimum completion time acceptable should be 60 seconds). These 41 answers were deleted before analysis. For the teacher questionnaire, 151 teachers completed it, and two of them in less than 45 seconds (with the same determining method mentioned above), which was deleted as invalid for further analysis. Therefore, a sample of 2310 first/second-year students and 149 college English teachers from 12 HEIs (three directly under MOE, six under Hubei Provincial Department of Education and three non-governmental, diverse in disciplinary areas, student enrollment numbers and include both research and teaching ones) was generated. The names of the institutions were de-identified during analysis for confidentiality reasons. The demographic information of the participants was presented in Tables 1 and 2.

## Data analysis

Quantitative data derived from the SRS and TRS were analyzed using SPSS 23.0 and Microsoft Excel. On the one hand, the numerical description of the variables was carried out (means, standard deviations) to understand the overall readiness of students and teachers for online college English learning/teaching during the online migration. On the other hand, comparisons of means with parametric and nonparametric tests were conducted to explore whether there were significant differences between different demographic groups.

Qualitative data obtained from open-ended questions were analyzed using topic and analytical coding [60]. Strict procedures were followed to ensure coding reliability. Firstly, the answers to the four questions were uploaded to the qualitative research program ATLAS. ti 8 as separate documents. Secondly, two researchers went through the documents to have a rough idea and started coding independently. The open coding and auto-coding functions were combined, but the auto-coding results were checked to avoid inappropriateness. Categories kept emerging through the process of coding. When this initial stage of independent coding finished, the two researchers compared their codes and categories to negotiate a final version. Lastly, categories were further analyzed.

## Findings

### Overview of college English education before and during COVID-19

From the interviews, an overall view about college English education in the sampled universities before and during the pandemic was obtained. The information was not directly related to the research questions, but it could help to better understand the statistical information in the subsequent sections.

Prior to the pandemic, five universities (three directly under MOE, two under provincial department of education) had implemented blended teaching with established online courses, four (under provincial department of education) was trying to integrate blended teaching or

**Table 1. Demographics for student respondents.**

| Variable | Category | n | % |
|---|---|---|---|
| Type of institution | Directly under MOE | 278 | 12.03 |
| | Under provincial department of education | 1076 | 46.58 |
| | Non-government HEIs | 956 | 41.39 |
| Disciplinary area | Philosophy | 5 | 0.22 |
| | Economics | 156 | 6.75 |
| | Law | 152 | 6.58 |
| | Education | 149 | 6.45 |
| | Literature | 129 | 5.58 |
| | History | 10 | 0.43 |
| | Science | 220 | 9.52 |
| | Engineering | 606 | 26.23 |
| | Agriculture | 24 | 1.04 |
| | Medicine | 181 | 7.84 |
| | Military | 2 | 0.09 |
| | Administration | 398 | 17.23 |
| | Art | 278 | 12.04 |
| Grade | First-year | 1388 | 60.09 |
| | Second-year | 922 | 39.91 |
| Gender | Male | 865 | 37.45 |
| | Female | 1445 | 62.55 |
| Area | City | 1332 | 57.66 |
| | Countryside | 978 | 42.34 |
| Device for online learning | Computer (desktop/ laptop) | 1160 | 50.22 |
| | Tablet | 81 | 3.51 |
| | Smartphone | 1069 | 46.27 |
| Type of internet connection | Broadband | 148 | 6.41 |
| | WIFI | 1775 | 76.84 |
| | 3G/ 4G/ 5G | 387 | 16.75 |
| Total | | 2310 | 100 |

students' online learning with classroom teaching, while the only three non-governmental institutions still relied heavily on traditional F2F teaching.

During the pandemic, all these 12 universities adopted a combination of synchronous and asynchronous activities with the help of existing online courses or newly-recorded lessons. To make teaching go smoothly against all kinds of adversities brough about by the pandemic, different kinds of technical tools were used. The mainstream ones were Chaoxing learning app and MOOC platform (40.16%), Icourse163 MOOC platform (28.78%), QQ (27.58%), WeChat (26.36%), VooVmeeting (22.16%), Tencent Classroom (21.30%), Dingtalk (18.10%) [61]. Teachers generally had their own choices with the tools by negotiating with students. Statistics showed that a teacher used an average of 2.16 platforms of tools, and only a minority of teachers (17.65%) managed to stick with only one platform or tool in teaching [61]. The platforms and tools share some common functions, but they have different strengths. Teachers tended to choose what they were familiar with at first and continued to combine other tools recommended by colleagues or students, especially when problems occurred. This led to the phenomenon of using a mixture of different tools.

In common practice, the final assessment for college English courses combines formative and summative assessments, with the former accounting for 30–50% focusing on assignments,

**Table 2. Demographics for teacher respondents.**

| Variable | Category | n | % |
|---|---|---|---|
| Type of institution | Directly under MOE | 48 | 32.21 |
| | Under provincial department of education | 75 | 50.33 |
| | Non-government HEIs | 26 | 17.46 |
| Gender | Male | 22 | 14.77 |
| | Female | 127 | 85.23 |
| Years in teaching | 0–5 years | 28 | 18.79 |
| | 6–10 years | 12 | 8.05 |
| | 11–15 years | 30 | 20.14 |
| | 16–20 years | 23 | 15.44 |
| | More than 20 years | 56 | 37.58 |
| Experience in online English teaching | Almost no | 83 | 55.7 |
| | Some | 56 | 37.58 |
| | A lot | 10 | 6.72 |
| Mode of delivery during pandemic semester | Synchronous | 68 | 45.64 |
| | Asynchronous | 13 | 8.72 |
| | Combination of the two | 68 | 45.64 |
| Total | | 149 | 100 |

online learning (if there is), class participation, attendance, and the latter 50–70% coming from the result of a final achievement test. When teaching was moved online during the pandemic for a whole semester, assessment became a big concern. Of the 12 universities interviewed, five maintained their former assessment criteria but moved achievement tests online, two delayed assessments according to institutional policy, four adjusted their criteria mainly in increasing the percentage of formative assessment and change question types for testing online. There was one university directly under MOE adopted a more complex approach on the basis of an investigation and analysis of learner characteristics. The assessment for first-year students remained unchanged but postponed while that for second-year students was changed into essay writing and group projects.

When asked about possible plans for online college English education, all personnel in charge of college English education claimed further integration of online teaching with F2F teaching and development of English MOOCs or intention to try blended teaching in the post-pandemic period if there were enough institutional support.

## Research question 1: To what extent were college students and teachers ready for online English education during the pandemic semester? Are there any individual differences?

**Student readiness.** Descriptive statistics from the SRS are presented in Table 3.

The overall level of readiness for online college English learning for the cohort of student respondents was 3.68 out of a score of 5. The highest scores were about technology access, mainly form the first three items. That is to say they had appropriate access to terminal devices with adequate hardware and software to study online during the pandemic semester. Item TA4 got a lower score in this dimension, but still high when compared with other items. This is due to the increasingly rapid development of information technology in China in recent years and the popularity of smartphones. As can be referred back to Table 1, 46.27% of the respondents had smartphones as their devices for online learning and more than three quarters of them used WIFI connection. It should be admitted that cases were reported as some

**Table 3. Means (M) and Standard Deviations (SD) of the SRS (N = 2310).**

| Dimensions & items | M | SD |
|---|---|---|
| *Technology access (TA)* | *4.03* | *0.95* |
| TA1 I have access to a computer/tablet/mobile phone. | 4.12 | 0.93 |
| TA2 I have access to a computer/tablet/mobile phone with adequate software (Office software, software for online learning, QQ) for my online learning. | 4.09 | 0.92 |
| TA3 I have access to internet connection (broadband/Wifi/3G/4G). | 4.13 | 0.96 |
| TA4 I have access to stable internet connection (fast, few failures) for my online learning. | 3.79 | 0.98 |
| *Computer/Internet self-efficacy (CIS)* | *3.69* | *0.93* |
| CIS1 I feel confident in performing the basic functions of Office programs (Word, Excel, and PowerPoint). | 3.58 | 0.95 |
| CIS2 I feel confident in my knowledge and skills of how to manage software for online English learning. | 3.66 | 0.94 |
| CIS3 I feel confident in using the Internet to find or gather information for English learning online. | 3.84 | 0.90 |
| *Self-directed learning (SDL)* | *3.48* | *0.91* |
| SDL1 I manage time well in English learning. | 3.38 | 0.92 |
| SDL2 I seek assistance when facing English learning problems. | 3.42 | 0.91 |
| SDL3 I carry out my own English study plan. | 3.47 | 0.92 |
| SDL4 I set up goals for my English learning. | 3.63 | 0.89 |
| SDL5 I have higher expectations for my English learning performance. | 3.49 | 0.90 |
| *Learner control (in an online context) (LC)* | *3.47* | *0.92* |
| LC1 I can direct my own English learning progress online. | 3.53 | 0.89 |
| LC2 I am not distracted by other online activities (online games, instant messages, Internet surfing) or distractions in my learning environment (TV, siblings playing) when learning English online. | 3.39 | 0.96 |
| LC3 I repeated the online instructional materials on the basis of my needs. | 3.48 | 0.90 |
| *Motivation for English learning (in an online context) (MFEL)* | *3.70* | *0.87* |
| MFEL1 I am open to new ideas. | 3.64 | 0.88 |
| MFEL2 I have motivation to learn English online. | 3.81 | 0.85 |
| MFEL3 I improve from my mistakes. | 3.74 | 0.86 |
| MFEL4 I like to share my ideas with others online. | 3.61 | 0.90 |
| *Online communication self-efficacy (OCS)* | *3.73* | *0.88* |
| OCS1 I feel confident in using online tools (email, discussion) to effectively communicate with others in English. | 3.79 | 0.88 |
| OCS2 I feel confident in expressing myself (emotions and humor) through texts in English. | 3.70 | 0.89 |
| OCS3 I feel confident in posting questions in online English discussions. | 3.71 | 0.88 |
| Overall | 3.68 | 0.94 |

All items were measured via a 5-point Likert scale: 1 = strongly disagree, 2 = disagree, 3 = neutral, 4 = agree, 5 = strongly agree

students from extremely underprivileged places or families had great difficulty managing to attend classes online but such cases were very rare among college students.

The respondents reported the lowest readiness score in self-directed learning and learner control dimensions, 3.48 and 3.47 respectively. Further analysis revealed that the lowest scores came from item SDL1 (*I manage time well in English learning.*) in self-directed learning dimension with an average score of 3.38 and item LC2 (*I am not distracted by other online activities— online games, instant messages—or distractions in my learning environment—TV, siblings playing—when having English class online.*) in learner control dimension with an average score of 3.39.

As for readiness in the rest three dimensions of computer/internet self-efficacy, motivation for learning and online communication self-efficacy, scores were around 3.7. Judging from the

total mean, it is possible to say that on average these student participants were moderately ready for online college English learning during the pandemic semester.

**Individual differences in student readiness.** Independent samples $t$-tests were conducted to explore the differences between different graders, genders and students from different areas in their overall level of readiness. No significant differences between different graders ($t$ = -0.75, $df$ = 1885.76, $p$ = 0.45) and genders ($t$ = 0.44, $df$ = 1499.51, $p$ = 0.66) were revealed. The independent samples $t$-test was significant ($t$ = 7.25, $df$ = 2280.27, $p<$0.01) on students from different areas as shown in Table 4. Students coming from cities rated higher scores in all the six dimensions of the SRS, especially in the dimensions of technology access and computer/internet self-efficacy, leading to a higher level of overall readiness (M = 3.78, SD = 0.78) than those coming from the countryside (M = 3.56, SD = 0.64).

Kruskal Wallis test showed statistically significant differences between groups from different types of institutions on levels of readiness ($H$(2) = 88.21, $p$ = 0.00) with a mean rank of 1385.53 (median = 3.95) for students studying in universities directly under MOE, 1223.10 (median = 3.77) for those from institutions under provincial department of education and 1012.53 (median = 3.55) for those from non-governmental institutions. Post hoc tests (Mann Whitney U tests) were performed to make pair-wise comparisons. Descriptive statistics showed that students from universities directly under MOE (median = 3.95, mean rank = 754.44) scored higher than those from institutions under provincial department of education (median = 3.77, mean rank = 657.62). Mann-Whitney $U$-value was found to be statistically significant $U$ = 128174.00 ($Z$ = -3.69), $p<$0.01, and the difference between the two groups was small ($r$ = 0.1) [62]. Students from universities directly under MOE (median = 3.95, mean rank = 770.58) also scored higher than those from non-governmental institutions (median = 3.55, mean rank = 572.98). Mann-Whitney $U$-value was found to be statistically significant $U$ = 90327.00 ($Z$ = -8.14), $p<$0.01, and the difference between them was a little below moderate ($r$ = 0.23) [62]. The last pair was students from institutions under provincial department of education and non-governmental institutions. The former group (median = 3.77, mean rank = 1103.98) scored higher than the latter (median = 3.55, mean rank = 918.04). Mann-Whitney $U$-value was found to be statistically significant $U$ = 420201.50 ($Z$ = -7.14), $p<$0.01, and the difference between them was a little below moderate ($r$ = 0.16) [62].

Kruskal Wallis test also revealed differences between groups using different terminal devices ($H$(2) = 104.50, $p$ = 0.00) with a mean rank of 1289.17 (median = 3.86) for students using computers, 1254.58 (median = 3.86) for those using tablets, and 1002.94 for those using smartphones. However, post hoc comparisons showed no statistical difference between students using computers and tablets ($p$ = 1.00).

**Table 4. Independent samples t-test of area.**

| | Area (Mean ± SD) | | $t$ | $df$ | $p$ |
|---|---|---|---|---|---|
| **Dimensions** | **Countryside (n = 978)** | **City (n = 1332)** | | | |
| Technology access | 3.90±0.84 | 4.13±0.88 | -6.43 | 2163.62 | 0.00** |
| Computer/Internet self-efficacy | 3.54±0.76 | 3.80±0.89 | -7.49 | 2254.30 | 0.00** |
| Self-directed learning | 3.35±0.71 | 3.57±0.87 | -6.61 | 2280.55 | 0.00** |
| Learner control | 3.35±0.75 | 3.55±0.89 | -5.82 | 2265.69 | 0.00** |
| Motivation for learning | 3.60±0.70 | 3.78±0.84 | -5.70 | 2268.26 | 0.00** |
| Online communication self-efficacy | 3.62±0.77 | 3.82±0.87 | -5.83 | 2230.32 | 0.00** |
| Overall | 3.56±0.64 | 3.78±0.78 | -7.24 | 2280.27 | 0.00** |

**$p<0.01.

In addition, Kruskal Wallis test revealed differences between groups using different internet connections ($H(2) = 74.01$, $p = 0.00$) with a mean rank of 1266.29 (median = 3.86) for the broadband group, 1203.76 (median = 3.77) for the WIFI group and 891.79 (median = 3.45) for the 3G/4G/5G group. Post hoc comparisons showed no statistical difference between students using computers and tablets ($p = 0.82$).

While existing studies rarely explored whether students of different disciplinary areas had different levels of readiness for online learning, this research took this aspect into account. Kruskal Wallis test revealed differences between groups with different disciplinary areas ($H(12) = 79.12$, $p = 0.00$). However, because of the limitation of the sample, some disciplinary areas had very limited number of respondents, making the results less convincing. To have a rough idea, the overall readiness levels were presented in Fig 1. If excluding areas in which the number of respondents were less than 100, students majoring in art, economics and engineering appeared to be readier than those in medicine and education.

**Teacher readiness.**   Descriptive statistics from the TRS are presented in Table 5.

The overall score of readiness for online college English teaching for the cohort of teacher respondents was 3.70 out of 5 based on the means of scores for three dimensions of technical readiness (3.73), lifestyle readiness (3.60) and pedagogical readiness (3.75). The teachers seemed to be more technically and pedagogically ready than being ready in lifestyle as getting access to and acquiring skills of using technical tools, and learning to teach online were less challenging than changing one's lifestyle at short notice when there were great emotional tensions during the pandemic. Though for item LR4, 79.86% of the respondents agreed or strongly agreed with the statement "*I value or need the flexibility of online teaching*", around 50% of them rated themselves as not "*having a private place to teach online* (item LR1)", being unable to "*work uninterruptedly for an extended period of time* (item LR2)" and not "*having people (family members, colleagues, friends or technical support from business) or resources (brochures or videos) to help them with technical problems* (item LR3)". Though the teachers reported higher scores of readiness in the pedagogical readiness dimension, a further probe exposed a low score in item PR6 "*I feel confident in engaging students in my online class*" (M = 3.57). Only 56% of the respondents showed confidence in engaging students in online class. This echoed with the findings in a previous study that 62.96% of online instructors had

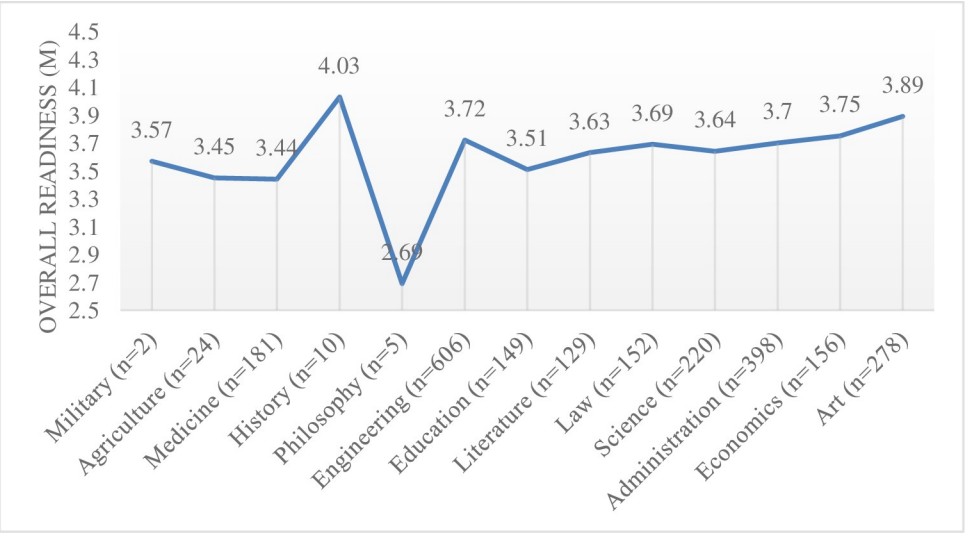

**Fig 1. Overall readiness (M) of students from different disciplinary areas.**

**Table 5. Means (M) and standard deviations (SD) of the TRS (N = 149).**

| Dimensions & items | M | SD |
|---|---|---|
| *Technical readiness (TR)* | *3.73* | *0.90* |
| TR1 My computer setup is sufficient for online English teaching. | 3.85 | 0.96 |
| TR2 My internet connection is stable for online English teaching. | 3.67 | 0.90 |
| TR3 I know how to use software and online teaching platforms to carry out and facilitate online English teaching. | 3.68 | 0.85 |
| *Lifestyle readiness (LR)* | *3.60* | *0.98* |
| LR1 I have a private place in my home or at work and that I can use for extended periods. | 3.60 | 1.00 |
| LR2 I have adequate time that will be uninterrupted in which I can work on my online courses. | 3.52 | 0.98 |
| LR3 I have people (family members, colleagues) and/or resources (manuals, videos) which will assist me with any technical problems I might have with my software applications as well as my computer hardware. | 3.30 | 1.08 |
| LR4 I value and/or need flexibility of online teaching. | 3.97 | 0.86 |
| *Pedagogical readiness (PR)* | *3.75* | *0.92* |
| PR1 When I am asked to use technologies that are new to me such as a learning management system or a piece of software, I am eager to try them. | 3.85 | 0.87 |
| PR2 I am a self-motivated, independent learner. | 3.78 | 0.90 |
| PR3 I can effectively design an online English class. | 3.68 | 0.87 |
| PR4 It is not necessary that I be in a traditional classroom environment in order to teach English. | 3.92 | 0.88 |
| PR5 I communicate with students effectively and comfortably online. | 3.74 | 0.92 |
| PR6 I feel confident in engaging students in online English teaching. | 3.57 | 0.95 |
| PR7 I feel comfortable checking students' assignments and providing different kinds of feedback online. | 3.72 | 0.99 |
| PR8 I have a positive attitude toward teaching and learning English online. | 3.72 | 0.94 |
| Overall | 3.70 | 0.94 |

All items were measured via a 5-point Likert scale: 1 = strongly disagree, 2 = disagree, 3 = neutral, 4 = agree, 5 = strongly agree

encountered the problem of poor involvement of students in online classes during the pandemic [63].

**Individual differences in teacher readiness.** An independent samples *t*-test revealed no significant difference between male teachers and female ones in their readiness levels ($p = 0.55$).

The ANOVA tests results were not statistically significant on types of institution ($p = 0.11$), levels of online teaching experience ($p = 0.95$) of the teacher respondents and different delivery modes ($p = 0.02$). Groups with different years in teaching didn't report different levels of readiness according to Kruskal Wallis test.

## Research question 2: What challenges did teachers and students encounter during the pandemic semester?

**Challenges from students' perspective.** For the open-ended question "What challenges have you encountered in your online English learning during this semester?", a total of 2310 answers were obtained, among which 1493 were meaningful and valid (The answers with only one word or meaningless signals were deemed invalid and deleted, while those using phrases or sentences were kept for analysis). Through careful and repeated reading, analysis and coding, six discernible categories of challenges reported by students emerged (Table 6).

The most frequently reported challenges were concerned with technical issues and different learning processes. For technical challenges, 224 out of 1493 respondents mentioned poor

**Table 6. Categories of challenges reported by students (codes and frequencies).**

| Technical challenges (427) | | Learning process (269) | | Learning environment (146) | | Self-control (139) | | Efficiency and effectiveness (123) | | Health concern (14) | |
|---|---|---|---|---|---|---|---|---|---|---|---|
| Internet connection | 224 | Communication | 98 | Distractions | 78 | Lack of self-discipline | 110 | Low efficiency | 72 | Eye fatigue | 14 |
| No textbook | 102 | Interaction | 67 | No learning atmosphere | 34 | Time management | 29 | Undesirable effectiveness | 51 | | |
| Devices for learning | 44 | Assignment | 39 | Low motivation | 28 | | | | | | |
| Apps or software | 37 | Examination | 20 | Lack of teacher presence | 6 | | | | | | |
| Cost of flow data | 12 | Speaking | 16 | | | | | | | | |
| Power failure | 8 | Note-taking | 15 | | | | | | | | |
| | | Listening | 14 | | | | | | | | |

internet connection and 102 thought learning without printed textbooks was difficult. There were also students having problems with their learning devices, apps or software and unexpected power failure, or complaining about their high cost of flow data. In terms of learning process, the biggest problem was difficult and unsmooth communication (Freq. = 98) as well as a lack of interaction with the teacher and peers (Freq. = 67). 39 respondents thought assignments for the online English course were heavier than before. Other challenges in this category included bad experiences in online examinations, inconvenience in taking notes during online classes and limitations in developing speaking and listening skills. Take one response for example: "*When learning English as a foreign language, we need to communicate with others orally and speak in front of the public as much as possible, but in online English classes we had neither opportunity.*" For learning environment and self-control, 110 respondents admitted their lack of self-discipline because they were invisible behind the screen and 78 were troubled by various kinds of distractions around. This corresponded with their low readiness score in learner-control dimension as analyzed in Section 4.2.1. One student said "*I was often called to do some housework and couldn't concentrate in class*". Another complained "*My little brother kept interrupting me when I was in class*". Participants also commented on the relatively low efficiency and undesirable effectiveness of online English classes compared with F2F ones. 14 students were worried about their worsening eyesight because of too much screen time, which was a problem not to be overlooked despite its low frequency.

**Challenges from teachers' perspective.** For the open-ended question "What challenges have you encountered in your online English teaching during this semester?", all the 149 answers from the teacher respondents were meaningful and kept for analysis. Three categories of challenges stood out after careful coding (Table 7).

The teachers' main difficulties during the online semester pertained to pedagogical issues, with a total number of 128 cases reported. The biggest concern was students' disengagement in online classes (Freq. = 58), followed by difficulties in tracking how well the students have learnt (Freq. = 37). 16 teachers mentioned that students' lack of self-discipline made teaching less effective; 12 thought shifting online had brought about more additional work; five were struggling with reviewing students' assignments online. More than one third of the teacher

**Table 7. Categories of challenges reported by teachers (codes and frequencies).**

| Pedagogical challenges (128) | | Technical challenges (52) | | Lifestyle challenges (6) | |
|---|---|---|---|---|---|
| Students' disengagement | 58 | Internet connection | 27 | Teaching environment | 4 |
| Tracking students' learning | 37 | Apps or software | 13 | Health concern | 2 |
| Students' lack of self-discipline | 16 | Devices for teaching | 12 | | |
| Workload | 12 | | | | |
| Students' assignments | 5 | | | | |

respondents encountered technical challenges, such as poor internet connection (Freq. = 27), difficulty in managing teaching software or apps (Freq. = 13) and low configuration of their devices for teaching (Freq. = 2). Except for pedagogical and technical challenges, a few cases were related to teaching environment and health concern. The following are some quotations from the answers:

> "The success of an online class largely depends on the students' self-discipline. Without teacher presence as in a F2F class, only those learners with self-discipline can learn well. Those who cannot control their own behaviors or keep themselves concentrated will drift away and lag behind."

> "The challenges involve unstable internet connection, platform's not being user-friendly and lack of comprehensive training for online teaching. In most cases, we are learning everything by ourselves from scratch and it is extremely time-and-energy-consuming."

> "There are abundant teaching platforms to choose from, each having its own advantages, but this makes teaching and learning more complicated and demanding."

> "How to engage students in online classes is a big headache because it's difficult to carry out class activities. Some students are too shy to turn on their microphone, especially in speaking tasks."

> "My laptop configuration is low and I have no private space at home for teaching. What's worse, I have to take care of my child and help with his online learning on weekdays too!"

## Research question 3: What were teachers' and students' perceptions towards future online English education?

From previous analysis, it was found that neither the students nor the teachers were perfectly ready for online English education. They also met a lot of problems during the process, some of which were specific to the pandemic context, and some were not. Do low readiness level and meeting problems lead to avoidance of online English education? The open-ended question "Are you willing to learn/teach English either in a fully online or blended course in the future? Why?" yielded useful information for reflection.

Among the 2102 students who gave valid responses to this question, 1144 expressed their willingness to learn English either in a fully or blended course in the future, 72 took a neutral attitude, 563 opposed the idea and the rest didn't show a clear position. Those who were in favor of online or blended English courses commented on the following five categories of advantages: flexibility, abundant resources, access to recorded classes, high efficiency and effectiveness. Meanwhile, those with opposing positions listed seven disadvantages: ineffectiveness, low efficiency, lack of self-discipline, lack of learning atmosphere, distractions, inconvenience, and being bad for eyes.

For the 149 teachers, 102 reported their willingness and plan to implement online or blended English teaching in the future because they had seen the complementary advantages of online teaching over F2F classes, experienced the flexibility it offered and its high efficiency. Some mentioned that going online was an overall trend in education that they were willing to and should follow. 17 had different attitudes for different situations as they would be happy to embrace the idea of online teaching in emergencies as we had because of COVID-19 but they still preferred F2F teaching. Only 16 claimed that they were unwilling to continue online teaching due to poor students' engagement, difficult class management and ineffectiveness. The rest were ambiguous in their attitudes without giving too many comments.

## Discussions and implications

This section will briefly summarize and explain the research findings listed above and draw implications for all relevant stakeholders in the process of online English education.

### Level of readiness and challenges encountered

According to Holsapple and Lee-Post [64], a student or a teacher can be considered e-ready with a mean score of 4 on a five-point Likert-type scale. As the overall means of the student respondents and the teacher respondents were 3.68 and 3.7 respectively, both cohorts were moderately ready, or rather, slightly below the ready level. Contrary to what one might expect, these digital natives could not be considered fully digitalized as proved by their lower score in computer/internet self-efficacy. This was also proved by Mehran et al [48], whose research found the digital natives in Osaka University had low digital literacy and competence for educational purposes. Moreover, students' unpreparedness was closely related to self-directed learning and learner control, which was consistent with previous studies [33, 65]. They displayed their unpreparedness not only through low scores, but also by reporting challenges about distracting learning environment, lack of self-discipline, and inability in time management. As Vonderwell and Savery [36] put it: self-directed learners know how to learn, how they learn, how to reflect on their learning, how to initiate learning and how to use time management skills efficiently. Mastery of these skills enable online learners to make efficient use of their time and resources available online. Agonács et al [66] proved that learners with higher self-directed learning readiness tend to have higher insight, self-efficacy and information navigation skills. Lian et al [56] suggested that students should develop both self-directed and collaborative learning skills in order to achieve meaningful learning in a technology-mediated authentic language learning environment both in or beyond the pandemic. Therefore, it is essential that instructors attach due importance to the cultivation of students' metacognitive skills related to self-directed learning, self-control or self-discipline and motivation, except for giving students orientation in technology aspects.

The analysis found no statistically significant difference in overall level of readiness between male and female students, which was proved by Hung et al [40]. No difference was found between freshmen and sophomore, either, which was inconsistent with the finding by Hung et al [40]. In their study different graders were found to have different levels of readiness. Contrastively, students from cities reported higher levels of readiness, not only in the overall scores, but in scores of every dimension. Despite the fact of overall development in China, it should not be neglected that gaps in development of all aspects including infrastructure, information technology, first and secondary education still exist between urban and suburban areas. It is highly possible that these gaps led to a difference in the readiness levels of students from different areas. Moreover, types of terminal devices and internet connection also seemed to be a variable impacting students' level of readiness. Students with better convenience, i.e. who had a computer or a tablet with broadband or WIFI connection were the readiest while those using smartphones connected by flow data appeared to be less ready. The mean scores were significantly higher for the advantageous groups especially in the technology access and computer/internet self-efficacy dimensions, proving stakeholders' technology access, and infrastructure may greatly impact what is possible [3]. The study failed to consider previous experience with online education as a variable, but students coming from different types of institutions were found to have different levels of readiness. The reason might be the fact that the three sampled non-governmental universities relied fully on traditional teaching for college English education while the others had already implemented blended teaching or online learning prior to the pandemic. There were statistically significant differences in overall readiness across different disciplinary areas. The reason may lie in the fact that for some disciplines, computer skills

is more important. Therefore, students from these disciplines basically had better computer and internet self-efficacy.

What drew our attention was students' opposing perceptions towards the efficiency and effectiveness of online English courses. Some students claimed that online English learning had advantages of high efficiency and effectiveness, while others emphasized its low efficiency and ineffectiveness. By looking at the respondents' scores of readiness and their answers to the open-ended questions, it was found that students who had lower scores of readiness in the self-directed learning, learner control and online communication self-efficacy dimensions generally considered online English courses less efficient and effective. These students usually lack self-discipline and can not regulate their own learning without the teacher's presence. As a result, they found they were benefiting less from the online English courses. Also, students who reported poor internet connection or software and hardware failures thought online learning less efficient. In large-scale online classes, problems may occur unexpectedly. And when they occur, it would take the teacher and the students longer time to address them. However, students who were capable of self-control and used to communicating in English online verbally or in written form viewed high efficiency and effectiveness as advantages of online English learning, because they could have their own pace of learning and have more chances to communicate with others. As for the cohort of teacher respondents, their low readiness level also needs great attention. Though they reported lower scores of technical readiness and lifestyle readiness, these problems are easier to be solved. In normal situations, staff members will have full access to all the school facilities. If the institutions give due importance to the infrastructure development, these problems will be addressed.

What need to be discussed were teachers' relatively high level of pedagogical readiness and the frequently reported pedagogical challenges. Teachers claimed to be positive towards online English teaching and were willing to learn new technology and skills. They also thought themselves capable of designing online English classes. However, during the pandemic semester, their primary concern was the outcome of teaching. In line with their low readiness score in students' engagement, the biggest challenge they met was how to effectively engage students in online classes. The problem was not only perceived by the teachers, but also reported by the students as bad communication and poor interaction. This problem was not unique to the current context, Maican and Cocoradă's [55] research also proved that the degree of participation and interaction was the most extensive theme reported by online foreign language learners during the pandemic in the context of Romania. Engagement, interaction, and communication can determine the success of an online course and the learner's learning performance [67, 68]. For English learning, engagement, participation, interaction or communication, whatever the name is, it is extremely important as Sun [20, 65] and Yang [68] emphasized that learner participation and interaction is in the central place and of crucial importance in successful language learning, whether it's face-to-face, blended or fully online teaching. Language learning is a skill-based process rather than a content-based one. Skill developments, such as the acquisition of speaking and listening skills, required constant synchronous interaction in the target language [69]. This was coherent with the students' responses referring to challenges in developing listening and speaking skills in online English classes. To solve the problem, efforts from both cohorts should be made. According to Moore's [70] interaction framework, there are three types of interaction in teaching and learning: learner-to-learner, learner-to-instructor and learner-to-content [67]. Therefore, teachers should delicately choose teaching contents and design activities that facilitate these types of interaction. Gacs et al. [3] also suggested using teaching platforms or tools that could support a communicative environment and taking advantage of authentic materials and multilingual online communities. Wang and Chen [69], and Aelterman et al [71] advocated real-time synchronous interaction in distance language

learning, which was used by about 90% of the respondents in the survey. For students' part, they need to be more motivated to participate in class activities and interact with peers and the instructor in order to practice English. The goal requires teachers' planning, guidance and encouragement, but most importantly, it can only be achieved through students' cooperation, as they are the center of teaching. For other challenges less frequently reported, such as assessment and evaluation, more factors should be taken into consideration in planning online or blended English courses.

## Implications for educational settings

Contrary to previous studies which summarized participants' unwillingness to take fully online or blended English courses [48, 72], only about 25% of the student respondents and 10% of the teacher respondents clearly opposed the idea of online or blended English courses, though they encountered and complained about so many challenges and rated themselves as slightly below the ready level. It is possible that despite all the difficulties, students and teachers who passively experienced online education in this unexpected situation still enjoyed and appreciated its advantages. As some respondents commented, moving online was also a trend of the times, and keeping up with the current was important for institutions, teachers and learners. It is necessary that all stakeholders in the educational setting work alongside to follow the trend and benefit from it.

Readiness is an important factor in the process of online language learning [27, 48, 67]. In order to ensure the success of an online English course, the prerequisite is to ensure that the students are ready for online learning. Forcing those who are not ready may cause them to have a negative view towards online English education, just as the survey has shown that 25% of the respondents who were forced to learn English online because of the pandemic clearly opposed the idea of online English learning. Measures should be taken to assess the students' readiness before designing and launching fully online courses or implementing blended teaching. Faculty members should provide students with sufficient training that can help them to improve their knowledge and skills relevant to the factors influencing readiness. Fan and Zou [73] emphasized the influence of both task designs and students' technology knowledge on the effectiveness of technology enhanced collaborative language learning. It is imperative that language teachers not only pay attention to students' technology skills, but pay more attention to pedagogical concerns. Because language teaching is very different from other courses, language teachers, when moving online, should be more cautious about their teaching approaches. They should design better course contents and activities in order to engage students and develop different language skills. What's more, different technologies can cater for the development of different language skills [73], so teachers should select appropriate technologies according to the focus of their courses and the target students. Experience is always gained through practice. English teachers should keep reviewing and reflecting on their practice and remain committed to change [20]. It is hoped that more English teachers can reflect on this forced and crisis-prompted online teaching experience and begin to get more training and more hands-on experience in developing fully online or blended courses.

Institutions are suggested to provide adequate infrastructure (including office areas, technological tools, efficient course management system and sufficient budgets) and additional training and support for teachers who want to develop online courses. They should also rethink the workload and evaluation issues of teachers who teach online or blended courses, because designing and implementing an online course usually is more time- and energy-consuming.

There are also IT companies who play a role in the development of online education. It is necessary that companies make online learning and teaching tools more responsive to

students' and teachers' needs, and enhance the interactivity of these tools. As using different tools in a single course will complicate the teaching and learning process, it is hoped that more comprehensive tools with a combination of different functions be developed. Companies should also provide clear instructions on how to integrate the tools in teaching and guarantee timely technical support for the users.

## Conclusion

This inquiry assessed student and teacher readiness for pandemic-prompted online college English education in Wuhan during the spring semester in 2020, explored challenges encountered by them and their perceptions towards future online English education.

For research question 1, both cohorts rated themselves as slightly below the ready level for this emergency migration online for English education. Individual differences exist between students from different areas and types of institutions, using different terminal devices and internet connection. Students from different disciplinary areas also showed different levels of readiness. For teachers, no statistical difference was revealed between different genders, types of institutions, experience in online teaching and years in teaching.

For research question 2, various categories of challenges were reported, some of which were related to personal skills and individual differences while some were more environment-bound. The students experienced more technical problems and challenges in the learning process, while the teachers were more frustrated with the problem of student engagement. Lacking self-discipline was also a prominent problem experienced by the students and sensed by the teachers.

For research question 3, having experienced a fully online college English course for a whole semester, a majority of the respondents showed their positive attitude toward online English education and claimed willingness to continue learning or teaching English in online or blended courses in the future. Implications for educational settings were provided based on the analysis of the research results.

There are three limitations associated with our research. First, despite the effort in covering respondents of a wider range, the samples were not evenly distributed and the teacher sample was not big enough. Second, the second question "Are you willing to learn/teach English either in a fully online or blended course in the future? Why?" included two concepts-"fully online course" and "blended course"-and didn't define the meaning of "future", which elicited more complicated results to be analyzed. Some respondents were actively commenting the two kinds of online courses and showing preference towards one of them, and some mentioned willingness to attention classes online in case of emergency but claimed unwillingness to do it as a new normal. Third, the group comparisons are not well theoretically based and the differences could be traced back to other third variables which was overlooked in the research. Based on this research, follow-up studies can explore the correlation between students' readiness level and their language learning outcomes, and how to engage students and develop a specific language skill in online English classes. How to assess and evaluate students' language learning outcomes online is also of great value. In addition, language teachers' professional development concerning online teaching can also be a possible research area.

## Supporting information

**S1 Table. Demographics for student respondents.**
(TIF)

**S2 Table. Demographics for teacher respondents.**
(TIF)

**S3 Table. Means (M) and Standard Deviations (SD) of the SRS (N = 2310).**
(TIF)

**S4 Table. Independent samples t-test of area.**
(TIF)

**S5 Table. Means (M) and Standard Deviations (SD) of the TRS (N = 149).**
(TIF)

**S6 Table. Categories of challenges reported by students (codes and frequencies).**
(TIF)

**S7 Table. Categories of challenges reported by teachers (codes and frequencies).**
(TIF)

**S1 Fig. Overall readiness (M) of students from different disciplinary areas.**
(TIF)

**S1 File. Questionnaire and interview results.**
(ZIP)

## Acknowledgments

We would like to thank all the teachers and students who participated in the survey and special gratitude goes to Professor Hung from National Chiao Tung University for sharing the Chinese and English versions of the OLRS.

## Author Contributions

**Conceptualization:** Ping Li.

**Data curation:** Li Jin.

**Investigation:** Cuiying Zou, Li Jin.

**Methodology:** Cuiying Zou.

**Project administration:** Ping Li.

**Resources:** Li Jin.

**Writing – original draft:** Cuiying Zou.

**Writing – review & editing:** Ping Li.

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
