## [Decision Letter · Decision Letter 0]

3 Jun 2021

PONE-D-21-14306

Online College English Education in China against the COVID-19 Pandemic: Student and Teacher Readiness, Challenges and Implications

PLOS ONE

Dear Dr. Jin,

Thank you for submitting your manuscript to PLOS ONE. After careful consideration, we feel that it has merit but does not fully meet PLOS ONE’s publication criteria as it currently stands. Therefore, we invite you to submit a revised version of the manuscript that addresses the points raised during the review process.

We look forward to receiving your revised manuscript.

Kind regards,

Di Zou

Academic Editor

PLOS ONE

Journal Requirements:

2. Please improve statistical reporting and refer to p-values as "p<.001" instead of "p=.000". Our statistical reporting guidelines are available at https://journals.plos.org/plosone/s/submission-guidelines#loc-statistical-reporting

Reviewers' comments:

Reviewer's Responses to Questions

**Comments to the Author**

1. Is the manuscript technically sound, and do the data support the conclusions?

Reviewer #1: No

Reviewer #2: Yes

Reviewer #3: No

Reviewer #4: Partly

Reviewer #5: Partly

Reviewer #6: Yes

Reviewer #7: Partly

Reviewer #8: Partly

Reviewer #9: Partly

Reviewer #10: Yes

2. Has the statistical analysis been performed appropriately and rigorously? 

Reviewer #1: No

Reviewer #2: N/A

Reviewer #3: No

Reviewer #4: Yes

Reviewer #5: N/A

Reviewer #6: Yes

Reviewer #7: No

Reviewer #8: I Don't Know

Reviewer #9: Yes

Reviewer #10: N/A

3. Have the authors made all data underlying the findings in their manuscript fully available?

Reviewer #1: Yes

Reviewer #2: Yes

Reviewer #3: No

Reviewer #4: Yes

Reviewer #5: No

Reviewer #6: Yes

Reviewer #7: No

Reviewer #8: Yes

Reviewer #9: Yes

Reviewer #10: Yes

4. Is the manuscript presented in an intelligible fashion and written in standard English?

Reviewer #1: No

Reviewer #2: No

Reviewer #3: No

Reviewer #4: Yes

Reviewer #5: Yes

Reviewer #6: Yes

Reviewer #7: No

Reviewer #8: No

Reviewer #9: Yes

Reviewer #10: Yes

5. Review Comments to the Author

Reviewer #1: This work describes students’ and teachers’ readiness towards online college English education. The authors have not published the results elsewhere. Although this study has presented the research into the welcome area concerned with online English education given the current circumstances, this study does not meet the required quality standards to be considered for publication.

The statistics and other analyses have mainly focused on students’ and teachers’ readiness towards online English education as an attitudinal factor. Although the investigation of readiness in attending online college English courses is certainly relevant given the growth of interest in this approach due to the COVID-19 pandemic, this study does not appear to have sufficiently contributed to the ongoing discussion and debate around this approach. The more important argument of this area as one of the blended learning approaches is the efficiency in English proficiency as the learning outcomes. The authors can present the literature and analyses concerned with the impacts of readiness on English proficiency in sufficient and well-substantiated detail.

When it comes to the discussions and conclusions, even if the authors have gained the conclusions are supported by the current data, the authors can extend the topic to make the discussions more comprehensive to some extent. For example, the authors can discuss the correlation between readiness and online English learning in terms of motivation, engagement, satisfaction, or cognitive load.

Considering the formats, although this study has met the community standards for data availability and the applicable standards for the ethics of experimentation and research integrity, the authors can pay attention to the latest APA guidelines that can provide valid references to an intelligible and standard fashion.

Thank you very much for the opportunity to review and consider your work.

Reviewer #2: Data acquisition is ambiguous. Questionnaires should be filled out in a relatively strict and specific environment. "By only including universities from which the interview was conducted 231 successfully within time", please specify this "time" duration.

Proofreading should be done.

Reviewer #3: The current study investigated virtual foreign language teaching and learning during the COVID pandemic by surveying students’ and teachers’ readiness. Through the distribution of a large-scale survey, the study collected a sizeable dataset to probe into the topic. The topic itself is valuable because of the utility of online language teaching. However, the study had serious methodological and analytical problems which precluded it from further consideration. I detail my comments below:

1. The theoretical framework section was actually the review of literature. There are a number of related theoretical underpinnings behind virtual teaching (e.g. computer assisted language learning, self-efficacy, teacher education, student motivations, etc). However, the paper lacks an in-depth discussion of how the empirical study was driven by theories.

2. The description of the measurements was far from clear. Sample questions of each dimension should be delineated in more detail.

3. The most critical question lies in the analytical tools. The description of data analysis including coding, reliability/validity checking, CFA and subsequent inferential statistics was insufficient. The methodology and data analysis were short on robust information. For instance, what does N refer to in the Table 3? If N indicates the number of each question type, the authors need to justify how 3 questions are sufficient to generate the meaningful pattern or factor loadings. Also, the CFA results were not properly presented.

4. The mixed-methods approach should have augmented the strengths of arguments. However, from the presentation of results, it remains unclear about how qualitative data supplemented the quantitative findings. How were the qualitative data sorted out and coded?

5. The data interpretation and discussion leads to the “so-what” concern. The researchers disseminated a large-scale survey to teachers and students across the region, which was commendable. Nonetheless, instead of the exploration of the status quo, the study should address how research findings can be converted to real pedagogical practices; and how instruction and learning can benefit from the empirical evidence.

Reviewer #4: According to the title, the authors claimed that it was the online College English Education “in China” that was investigated. However, only higher education institutions in Wuhan were examined. Therefore, I believe that the authors need to consider whether data gained from the survey conducted in Wuhan were representative enough to represent the condition of online English education in the whole nation (which refers to China). This deficiency provides the rationale for my recommendation to be “major revision.”

The first sentence of the abstract “China was the first country to migrate almost all teaching, learning and even assessment online in education of all levels against COVID-19” (Line 12, 13). The authors failed to provide any proof of this claim, neither in the Abstract nor in the main body of this article. I believe that it is important to explain this statement.

Some unidiomatic word usage is noticed “it was not based on nowhere” (Line 37), “even unimagined” (Line 50), and what HEI (Line 40, where this abbreviation appears for the first time) refers to is not known. The author needs more grammatical checks of the entire article.

The second section should be renamed as “Literature Review” rather than “Theoretical Framework,” considering that most of the discussions in this section were related to the results of the previous research. This means that the authors may need to rewrite the whole section to demonstrate their theoretical foundation. Some introductions to online language education in section 2.1 should have been offered in the first section (Introduction).

Reviewer #5: 1. No full name of HEI and SPOC was given.

2. The number of “3.5 Survey dissemination” should be 3.4.

3. The format of the two headings 4.1& 4.2 is inconsistent.

Theoretical Framework：

4. L117 (Line117): Quote format error: names of more than three authors should be cited as “et al.”

5. L126: How is language teaching different from other subjects?

The research gap that “Language learning online is different from online learning of other subjects, but until now, few studies have focused on student readiness specifically for online language learning” is not fully discussed in terms of what are the difference of language online learning and other subjects online learning”

6. What is your research gap? Is “studies on student readiness on online learning” or “student readiness specifically on language online learning”?

7. Previously the author mentioned “Major dimensions included in these scales are student access to/use of technology, online skills and relationships, motivation and interest, self-directed learning, learner control, factors that affect success”, but on page 10, the dimension of your questionnaire are almost the same as previous researches.

8. L76: There are differences between pandemic and epidemic. It should be “COVID-19 pandemic”.

9. L85-89: grammatical mistakes. "It is flexible; can be adaptive; allows for enhanced, individualized, and authentic materials; can take advantage of communicative tasks and multilingual communities; can foster and take advantage of autonomous learning and learner corpora"

10. L103 "Many HELs are developing MOOCs and SPOCs for English education online, but are teachers and students ready for the development? Literature in this domain was relatively scare." Before addressing the research gap, the authors mentioned several literature on online education. However, there are literature related to MOOC for English learning, maybe the authors could add more literature related to online language learning.

Methodology：

11. Why there are only student respondents from first- and second-year grade?

12. The specific procedure in which the researchers used the tool, ATLAS.TI 8, for coding needs elaboration.

13. Please make sure that the sentence “Qualitative data derived from the SRS and TRS were analyzed... as well as establishing...” is grammatically correct.

14. L229: it is not clear that 18 refers to what here?

15. L293-295: there are no statistic supports for the mainstream platform like Tencent Classroom.

Findings：

16. L524-526: the authors mentioned that “perceptions for future online English education, some students think online or blended English courses have the advantage of high efficiency and effectiveness”. But in line479-481, “participants also commented on the relatively low efficiency and undesirable effectiveness of the online English classes compared with F2F ones”. Further analysis of what led to this result is needed.

17. In 4.1 the authors mentioned about college English education before, during and after COVID-19. But how to define “after COVID-19”? The overview of college English education before and during COVID-19 was supported by data, but overview of college English education after COVID-19 was not.

18. L434-436: Why echo with an unpublished paper? 4.3.2 and 4.3.3 are from two open-ended questions. What about the data of interview?

19. Concerning the finding in p22 “Teachers generally had their own choices with the tools by negotiating with students but a common phenomenon was a mixture of different tools”. The author could further discuss more about the phenomenon of mix use of different tools. Or if it was from the previous studies, the authors should give clear citations.

Discussion & implication：

20. L526-529: no discussion for why students’ perception of online courses varies so differently. Some students view online learning has advantages high efficiency and effectiveness. While other student consider online learning has disadvantages: ineffectiveness, low efficiency.

21. Line 578: why the authors said "with the exception of pedagogical readiness...", while in page 23, especially in item, pedagogical readiness got a low score. It would be better to discuss the low score of item13.

22. L599-602 The expresssion seems confusing and could be polished.

Reviewer #6: Overall, the study is well designed and well researched. The author chose online college English education in China during COVID-19 pandemic as the research topic to investigate students’ and teachers’ readiness and challenges through different dimensions. The samples in this study were mainly collected from one province, which might be one of the limitations of the study. As for the methodology adopted in this study, although the author used factor analysis to extract several different dimensions (factors) behind items in questionnaire, the author did not make further explanations about those different factors and most of the statistics were conducted to make comparisons between different variables. I think the author should add some explanations to illustrate those different factors before making further comparisons between different variables.

The paper is generally written in quality academic style but there are several language issues worthy of revision. Please note that page numbers reflect the page number as shown in my pdf reader and do not reflect actual page number of the manuscript. I list several several issues below (though it is not intended to be exhaustive):

P3 HEI ---Higher Education Institutions (HEIs)

Mooc---Massive Online Open Course(Mooc)

P5 utilizes---utilize

“It is flexible; can be adaptive; allows for enhanced, individualized, and authentic materials; can take advantage of communicative tasks and multilingual communities; can foster and take advantage of autonomous learning and learner corpora….”

The author used semicolon after each clause. However, most of the clauses lack subjects. If the author insists on using such semicolon structure, I suggest the author to add subjects to those clauses. Perhaps another choice is to rearrange the structure of the sentence to make it more integrated.

P7 student access ----student’s access

was----is?

“...teachers need different skills than those normally employed by tutors trained to teach languages in a F2F classroom…..”

This sounds confusing. I think the author might consider add “and” between “tutors” and “trained”.

P8 lower level skills as --- lower level skills such as ?

higher level skills as—higher level skills such as ?

P13 “The survey lasted for ten days.” I suggest the author to combine this sentence with the previous paragraph instead of leaving it as a separate paragraph.

P14 “As the SRS and TRS were…” I suggest the author to change “as” into “Since”

P15 “a combine of” ----a combination of

P19 “suburb area”---suburban area

P20 “test were”---test was

P36 “ ....teachers who have taught online and are planning to teach online or blended English courses.” --- The sentence is ungrammatical.

“.... businesses which invest in education technology and want to draw more profits.” --- The sentence is also ungrammatical.

Some other points:

1.Page 4: The author mentioned that online English course is different from online courses in other disciplines. Perhaps the author should provide some explanations to illustrate the reason why online English teaching is different and much more difficult to implement.

2. P6 “Literature in this domain was relatively scarce.” This sounds a little incomplete when the author put it at the end of the paragraph.

3.“Section 4.1 overview of college English education before, during and after COVID-19”

In this part, the author listed the different dimensions derived from the questionnaires. However, it seems that the author didn’t make an explicit explanation about those dimensions listed in table 3.

Reviewer #7: The paper reported a survey to evaluate the readiness for online English education among 2310 non-English majors and 149 teachers in China. The results revealed a relatively low level of readiness among students and teachers. The study also identified the challenges participants faced in online language education. I appreciate the time and efforts the authors have put into this work, but I do have a number of major concerns which are detailed below.

1. Although the title of the manuscript is “Online College English Education in China against the COVID-19 Pandemic”, the data were collected from one single city in China. It’s hard to say that they can represent the entire country.

2. Three research questions were listed in the paper, but their relationship is rather unclear. For example, if participants report a lower level of readiness at certain dimension, it is likely for them to report greater challenges at that dimension. So the first two questions are to a certain extent overlapping. I suggest the authors reconsider the research questions to avoid overlapping, and if there is any change in the research questions, the entire paper needs to be updated accordingly. It is also necessary to explain clearly why the third research question needs to be explored.

3. Although the authors indicate that “All relevant data are within the manuscript and its Supporting Information files” in the Data Availability Statement, I cannot find the data in the manuscript, and it seems no relevant supporting files have been provided. PLOS ONE requires the authors to share their data publicly, but right now I can only find the statistical results which are not the same as the data. Therefore, I strongly recommend the authors to upload their source data.

4. The abstract fails to summarize the key findings from the research. For instance, what are the challenges encountered by the students and teachers?

5. P2 line22-23: “some challenges coherent with their low scores in certain dimensions from the readiness scales”

Since the authors did not mention what the challenges entail, there is no way for readers to understand the meaning of the sentence.

6. P2 line 24-27: “Qualitative data also showed prospects of growing development of online college English education as the majority of respondents reported their willingness and intention to continue learning/teaching English in online or blended courses in the post-pandemic period.”

This conclusion is not convincing. It is oversimplified to conclude that online college English education has potential to grow only on the basis of the fact that the respondents reported their willingness and intention.

7. If the participants were anonymously engaged in the survey, and they were not compensated in any way, how can the authors ensure they were sufficiently motivated to provide genuine answers to the questions? In designing the scales, did the authors include any item to detect lies? Have the quantitative data been filtered in any way before the analysis?

8. P6, line 105: “Literature in this domain was relatively scarce.”

Does this indicate there is no literature on this topic at all, or there are only a small number of them? Please make it clear.

9. P7 line 126: “few studies have focused on student readiness specifically for online language learning.”

But there are indeed a number of studies on student readiness for online language learning (e.g., Tylor &David 2013; Luu & Lian, 2020; Mehran, et al., 2017). I suggest the authors review the prior studies in a more comprehensive way. The relevant studies should be cited. More importantly, the authors should specify what new contribution this study has made to this field, given all the previous studies that have already been conducted. In reviewing the prior studies concerning teacher readiness, the authors also need to point out the inadequacy of these studies had or the issues they failed to address, which make it necessary for the authors to carry out their own study.

10. The following methodological concerns need to be addressed in the study:

What is the purpose of combining qualitative and quantitative approaches in this study?

What methodological framework did the study follow?

11. A semi-structured interview seems to be conducted in this study, but the findings from the interview do not seem to be relevant to the research questions the authors intended to address. The authors need to reconsider the purpose of having a semi-structured interview? If it is necessary to have the interview, the relevant details need to be reported: How was it designed? What procedures were followed? Did the participants answer any question? How were the answers analyzed?

12. The data analysis section is not helpful for readers to understand how the data analysis was made. The authors only introduced the tools for statistical analysis, which is far from enough. They should also report the statistical approaches used in the analysis, how data trimming was performed, the procedures of data analysis, how the interview was transcribed, how they ensured the accuracy of transcription, etc. The statements about research instrument should be moved to the Instrument section.

13. The results section was not written in a logical way. I suggest the authors reorganize it to respond to the three research questions. Section 4.1 does not seem to be relevant to any research question. Also the relation between 4.2.1 and 4.2.2 is not clear.

14. The statistical results were not reported in a standard way. For all the results from t test or ANOVA, degree of freedom and effect size should be reported. For any insignificant results (e.g., line440-441), the authors should report the statistical results with the exact p values. All statistical symbols (sample statistics) that are not Greek letters should be italicized.

15. The quantitative study shows that students were most ready in terms of technology access, which seems to contradict the finding from the qualitative study which indicates that the greatest challenge for students was technical. Does this divergence reflect an inadequacy in the design of the student readiness questionnaire? More explanation is needed.

16. P27 line 489-491: Despite the teachers’ higher scores of pedagogical readiness in the TRS, their main difficulties during the online semester pertained to pedagogical issues…

Please account for the discrepancy.

17. In the Results section, the authors have reported the results concerning the individual differences in students or teachers, but very few of them were summarized and discussed in the Discussion section. The discussion is rather inadequate.

18. The language of the manuscript is below the standard for publication. It contains too many grammatical and other errors, and some sentences are very difficult to understand. It needs to be proofread thoroughly and carefully by a native speaker of English. Below I’ve listed some of the problems, but such problems are almost everywhere:

P5 86-89: “It is flexible; can be adaptive; allows for enhanced, individualized, and authentic materials; can take advantage of communicative tasks and multilingual communities; can foster and take advantage of autonomous learning and learner corpora”

The sentences are fragmented.

P6 line98: in infancy level > at its infancy

P6 line113: either fully or hybrid > either fully or hybridly

P7 line129: Concerning instructors, teaching online learning requires a reconstruction of their…

What does “teaching online learning” mean?

Line550: understanding how was the situation>understanding the situation

Line 613: there are 3 types interaction

Reviewer #8: PONE-D-21-14306 comments

The study investigated the student and teacher readiness of online teaching and learning in the Chinese EFL contexts. Due to the following methodological issues, I may not believe the article is ready for publicaiton in the Journal in its present form.

1. pp.10-11 "The two questionnaires were initially piloted to check clarity of the language used and to ensure the reliability and validity of the two scales in the local context. Improvements were made in light of the comments from pilot respondents and two experts in research methodology. Both scales were statistically reliable and valid with the pilot tests."

-- Could you please describe in more detail what "improvements" or modifications you did on the questionnaires? In addition, what are the results of the statistical analyses for the evaluation of the reliability and validity of the questionnaires in the pilot tests?

2. p.11 "a semi-structured interview was conducted between the researchers and the personnel in charge of college English education from the sampled universities or colleges to get an overall view of the situation before, during and after the pandemic semester for the sake of better understanding of the statistical results of the survey."

-- Could you describe in detail how the interview was administered? In particular, how did you choose the participants of the interview? What was the procedure of the interview?

3. p.14 "Cronbach’s alpha coefficients were calculated to ensure internal consistency and confirmatory factor analysis was carried out to provide evidence for convergent validity."

-- The readers may also be curious of other results of the CFA than those you reported in Table 3. Particularly, what is the overall result/significance of the model?

Reviewer #9: This study investigated the readiness of students and teachers for online college English education in the spring semester of 2020 in Wuhan during Covid-19, and examined the challenges they faced and their perceptions of the future of online English education.

I have some general comments which hopefully the authors can consider when revising the manuscript.

Introduction and theoretical framework:

1. On page 4. The author mentioned that “…lessons can be and should be drawn for future development of online college English education”. It is necessary for the authors to clarify why and how the examination of emergency remote learning that occurred during Covid-19 can hold implications for that in the non-pandemic periods. Remote learning during a pandemic has characteristics that are not present during non-pandemic periods.

2. In the section titled theoretical framework, it seems that the authors did not elucidate clearly or even did not elucidate any theories behind this study. The authors need to consider introducing a theory (theories) supporting this study.

3. The empirical studies reviewed are not those conducted during the period of Covid-19. The authors need to consider focusing the literature review on or most of the reviewed should be from the pandemic period.

4. On page 6. The author mentioned “…but are teachers and students ready for this development (developing MOOCs and SPOCs)?” I think there are some studies exploring student and teacher readiness for MOOCs and SPOCs. Also, there is a difference between readiness for MOOCs and that for emergency remote learning. Authors should focus on reviewing studies that examined remote learning during emergency situations.

Methodology:

1. Authors need to clarify how the questionnaires were translated into Chinese?

2. Authors need to explain how the data were filtered and how invalid data were removed.

3. What is the purpose of conducting confirmatory factor analysis on both pilot and final sample? Typically, we use the pilot sample for exploratory factor analysis and the main sample for confirmatory factor analysis.

Findings:

1. Section 4.1 could be written more concisely. And it can be considered in the methodology section as research background.

2. The results would be more informative if demographic characteristics were in a regression model as predictor variables to assess their predictive power on each indicator of readiness. It seems that prior experiences with online education should be an important factor influencing readiness. Did the authors consider this factor?

3. The authors examined whether there were significant differences in overall readiness by grade and gender, but what about each indicator of readiness?

4. The authors examined differences in readiness for online education across disciplines. However, as I understand it, the authors wanted to examine the readiness of students for online English education. So why is it important to examine readiness for English online classes across disciplinary backgrounds?

5. The authors examined both student and teacher readiness. And what is the connection between them? Why is it important to discuss both the teachers’ and the students’ readiness in one article if the connection between was not explicitly examined?

6. The authors used interviews to examine the challenges students faced. I wondered why the purpose of the interviews was not to examine the factors that influence success and failure in readiness for Covid-19 emergency remote learning. This would make the article more focused.

7. The research question addressed in section 4.3.3 was “Are you willing to learn/teach English …in the future?” How would authors define the future? The future with a pandemic or emergency situations? Or a future without any emergency situations?

Discussion

1. The discussion needs to be deeply integrated with the relevant literature conducted during Covid-19.

2. The pedagogical implications should be stated more specifically and concretely.

Overall, the authors have a large sample size and the findings show the preparedness of students and teachers during the Covid-19 outbreak. But the entire article (literature review, findings, discussion) needs to be more focused, and closely aligned with the context of the study during the pandemic.

Reviewer #10: Recommendation: Revisions Required

The paper investigated 2310 non-English-major students and 149 English teachers from three types of twelve higher education institutions in Wuhan, to evaluate their readiness for online English education, to figure out challenges they encountered, and to draw implications for future online college English education. It has various potentials and I recommend it to be considered for publication, but only after some major revisions. My suggestions are the following:

1.The authors failed to evaluate the context of the nation-wide online teaching in a justifiable way. They claimed that “This online teaching in the face of COVID-19, or exactly, online triage (Gacs et al. 2020), was carried out without need analysis or readiness evaluation from both learning and teaching sides and was different from well-prepared and planned online teaching.” But actually it was not the case: before the epidemic, online teaching had already serve as a positive supplement to offline teaching, and it is available and accessible for students in most part of China, both in rural and urban areas; there are many university websites and apps which offer free online courses, which quite a number of students make use of regularly in their spare time; great efforts have been made by education authorities, especially the Ministry of Education, to analysis and evaluate the situation, to invest considerably in providing online teaching resources, and to mobilize the education-related IT companies to give a helping hand; schools and teachers had tried their best to make preparations before the launch of the online courses. In short, this nationwide online teaching experience was not unprepared and unplanned at all. So, the authors should adjust their wording throughout the whole paper, in order not to make the basis of this research seemingly groundless.

2.Get the paper checked by expert speakers. The sentences are extremely long, which makes reading the text challenging. And occasionally, there are some inappropriate wording or even grammatical mistakes. All these may reduce the readability of this paper to some extent.

3.The teacher sample is relatively small, which makes the analysis and results less convincing.

4.The analysis would be more reasonable and sound, if the subjects of teachers and students were divided into two groups, rural and urban.

5. It would be better to make the review of literature more pertinent to online English teaching. In the present form, this paper focuses too generally on online teaching as a whole. Thus it is advisable for the authors to make a very brief review of general online teaching but to concentrate more on English teaching, just as the authors claim in the paper that online teaching is carried out differently to cater for the need of different subjects or disciplines.

6. PLOS authors have the option to publish the peer review history of their article (what does this mean?). If published, this will include your full peer review and any attached files.

Reviewer #1: No

Reviewer #2: No

Reviewer #3: **Yes: **Haomin Zhang

Reviewer #4: No

Reviewer #5: **Yes: **Xiaobin Liu

Reviewer #6: No

Reviewer #7: No

Reviewer #8: No

Reviewer #9: No

Reviewer #10: No

---

## [Author Response · Author response to Decision Letter 0]

30 Jul 2021

Response to Reviewer #1

Thank you very much for your careful review and valuable suggestions. Our response and improvement are as follows.

1. The statistics and other analyses have mainly focused on students’ and teachers’ readiness towards online English education as an attitudinal factor. Although the investigation of readiness in attending online college English courses is certainly relevant given the growth of interest in this approach due to the COVID-19 pandemic, this study does not appear to have sufficiently contributed to the ongoing discussion and debate around this approach. The more important argument of this area as one of the blended learning approaches is the efficiency in English proficiency as the learning outcomes. The authors can present the literature and analyses concerned with the impacts of readiness on English proficiency in sufficient and well-substantiated detail.

We were aware of the limitations of the research, but we also believed in its contribution to fully online college English teaching in emergencies and blended teaching as common practices, especially in China, where online English teaching has become a commonplace but online English learning readiness is not well researched. Thank you for giving us inspiration for our future research in this area. Based on this study, we are designing a study on the correlation between online English learning readiness and students’ satisfaction and their English proficiency. 

2. When it comes to the discussions and conclusions, even if the authors have gained the conclusions are supported by the current data, the authors can extend the topic to make the discussions more comprehensive to some extent. For example, the authors can discuss the correlation between readiness and online English learning in terms of motivation, engagement, satisfaction, or cognitive load.

Thank you for your suggestions. The discussion part has been improved, but for extending the topic, we are sure to take your suggestions into consideration when designing new studies in the future.

3. Considering the formats, although this study has met the community standards for data availability and the applicable standards for the ethics of experimentation and research integrity, the authors can pay attention to the latest APA guidelines that can provide valid references to an intelligible and standard fashion.

The format has been modified to meet the guidelines of PLOS ONE. 

 

Response to Reviewer #2

Thank you very much for your review and valuable suggestions. We have revised the manuscript based on your comments.

1. Data acquisition is ambiguous. Questionnaires should be filled out in a relatively strict and specific environment. “By only including universities from which the interview was conducted 231 successfully within time”, please specify this “time” duration.

The data acquisition process has been clarified. We combined the original “3.3 Participant recruitment” and “3.4 Survey dissemination”, retitled the section as “3.3 Data acquisition”, and changed some wording (e.g. “time” duration mentioned in your comments) to make it concise, specific and understandable. 

We appreciate your advice for a strict and specific environment to fill out the questionnaires. However, universities in Wuhan had not been reopened when the survey was carried out from July 24th to August 2nd. Moreover, we wanted to carry out the survey immediately after the online semester was terminated. Therefore, we had no choice but to carry out the survey through online platforms instead of using paper questionnaires in the classroom with a teacher’s presence. We will pay special attention to data acquisition in future research.

2. Proofreading should be done.

Proofreading has been done to polish the language and modify the format.

 

Response to Reviewer #3

Thank you very much for your review and pointing out the problems concerning methodology and data analysis. We have revised the manuscript based on your comments.

1. The theoretical framework section was actually the review of literature. There are a number of related theoretical underpinnings behind virtual teaching (e.g. computer assisted language learning, self-efficacy, teacher education, student motivations, etc). However, the paper lacks an in-depth discussion of how the empirical study was driven by theories.

Thank you for your reminding. The section was retitled as literature review, and in this part some discussions about TAM and major components of a readiness scale were added to provide theoretical background for the study.

2. The description of the measurements was far from clear. Sample questions of each dimension should be delineated in more detail.

The instrument section was improved. And in the findings section, we have provided the detailed items of the whole scale.

3. The most critical question lies in the analytical tools. The description of data analysis including coding, reliability/validity checking, CFA and subsequent inferential statistics was insufficient. The methodology and data analysis were short on robust information. For instance, what does N refer to in the Table 3? If N indicates the number of each question type, the authors need to justify how 3 questions are sufficient to generate the meaningful pattern or factor loadings. Also, the CFA results were not properly presented.

Thank you for telling us even the smallest detail of inadequacy in our description. We have double checked our analysis and improved the description. 

After careful consideration, we decided to exclude details about CFA. We used scales validated in previous studies with pre-determined dimensions, and the focus of the study was not to validate the scales, so we didn’t carry out EFA with the pilot sample. We intended to show details of CFA with the main sample, and the KMO values indicated that the samples were adequate for CFA, but the final results were problematic. That’s why we only included information of AVE and CR in the previous Table 3. However, we still wanted to show the results of the survey, so we changed our way of analysis for a little bit and exclude details about CFA. We are sorry for this limitation, and we will be more conscientious with future studies.

4. The mixed-methods approach should have augmented the strengths of arguments. However, from the presentation of results, it remains unclear about how qualitative data supplemented the quantitative findings. How were the qualitative data sorted out and coded?

The qualitative data help better explain the participants’ readiness scores. Information about how were the qualitative data sorted out and coded were added in the new sections 3.3 data acquisition and 3.4 data analysis.

5. The data interpretation and discussion leads to the “so-what” concern. The researchers disseminated a large-scale survey to teachers and students across the region, which was commendable. Nonetheless, instead of the exploration of the status quo, the study should address how research findings can be converted to real pedagogical practices; and how instruction and learning can benefit from the empirical evidence.

We intended to reveal the status quo through the survey, but we failed to detail the pedagogical implications. We tried to improve the discussion part to explain the implications in the education setting. 

 

Response to Reviewer #4

Thank you very much for your review and constructive suggestions to improve our manuscript. We have revised the manuscript based on your comments.

1. According to the title, the authors claimed that it was the online College English Education “in China” that was investigated. However, only higher education institutions in Wuhan were examined. Therefore, I believe that the authors need to consider whether data gained from the survey conducted in Wuhan were representative enough to represent the condition of online English education in the whole nation (which refers to China).

Initially, we believed the data gained from the survey conducted in Wuhan were representative enough to represent the condition of online English education in China for the following reasons. Firstly, according to the statistics released by the Ministry of Education (MOE), Wuhan (the capital city of Hubei Province) has the largest student population and the second biggest number of HEIs in China. Secondly, college English education in China follows similar patterns guided by the College English Curriculum Requirements released by the MOE. 

However, after careful consideration based on your comments, we decided to change “in China” into “in Wuhan”. Our reasons are: 1) Though college students in Wuhan come from all over the country, the majority are from Hubei Province because of the college student recruiting policy; 2) There are differences concerning technological development between different cities. 

2. The first sentence of the abstract “China was the first country to migrate almost all teaching, learning and even assessment online in education of all levels against COVID-19” (Line 12, 13). The authors failed to provide any proof of this claim, neither in the Abstract nor in the main body of this article. I believe that it is important to explain this statement.

According to the prevention measures against the spread of COVID-19 on campuses issued by the MOE, all schools were closed. HEIs were asked to organize online teaching and local education authorities were asked to mobilize resources to provide online courses to secondary and primary school students. The first mass outbreak of COVID-19 was in China, so the sentence was written in this way. Thanks for your reminding, we have rewritten the sentence as “China migrated all teaching and learning online in education of all levels against the spread of COVID-19 on campuses (MOE document, 2020)”. 

3. Some unidiomatic word usage is noticed “it was not based on nowhere” (Line 37), “even unimagined” (Line 50), and what HEI (Line 40, where this abbreviation appears for the first time) refers to is not known. The author needs more grammatical checks of the entire article.

Thank you for your careful checks. The mistakes mentioned have been corrected and the language of the whole manuscript has been polished.

4. The second section should be renamed as “Literature Review” rather than “Theoretical Framework,” considering that most of the discussions in this section were related to the results of the previous research. This means that the authors may need to rewrite the whole section to demonstrate their theoretical foundation. Some introductions to online language education in section 2.1 should have been offered in the first section (Introduction). 

The second section was renamed as “Literature Review”. The Previous 2.1 about online language education in general was moved to the introduction part, and the whole section was rewritten and some relevant information as theoretical foundation (such as TAM, definitions of the components mentioned in the readiness scales) was added. 

 

Response to Reviewer #5

Thank you very much for your careful review and valuable suggestions to improve our manuscript. We have revised the manuscript based on your comments.

1. No full name of HEI and SPOC was given.

The full names of HEI and SPOC were added when they appear for the first time in the manuscript.

2. The number of “3.5 Survey dissemination” should be 3.4.

The numbers of the sections have been rearranged.

3. The format of the two headings 4.1& 4.2 is inconsistent.

The format of the manuscript has been revised to be consistent with the journal.

Theoretical Framework：

4. L117 (Line117): Quote format error: names of more than three authors should be cited as “et al.”

The quote format of the manuscript has been revised to be consistent with the journal.

5. L126: How is language teaching different from other subjects?

The research gap that “Language learning online is different from online learning of other subjects, but until now, few studies have focused on student readiness specifically for online language learning” is not fully discussed in terms of what are the difference of language online learning and other subjects online learning”.

Further explanation was added: Online language learning is different from online learning of other subjects. Unlike other subjects, language is both the means and the ends of online learning. In the learning process, the learners are supposed to listen, speak, read and write in the language they are learning. Therefore, whether the online learning environment can provide opportunities for the learners to use the language and whether the learners feel free to use it online determines the success of the language course. While studies on readiness prevail for general online learning, there seems to be only a few focusing on online language learning.

6. What is your research gap? Is “studies on student readiness on online learning” or “student readiness specifically on language online learning”? 

We have revised the literature review part and further reviewed several studies on learner readiness on online language and summarized our research gap: To the best of the researchers’ knowledge, there has been no study conducted to assess Chinese college students’ readiness for online English learning. Fully online teaching for college English was implemented because of the pandemic, and the researchers took this opportunity to carry out the study, in order to have a rough idea about college students’ readiness for online English learning.

7. Previously the author mentioned “Major dimensions included in these scales are student access to/use of technology, online skills and relationships, motivation and interest, self-directed learning, learner control, factors that affect success”, but on page 10, the dimension of your questionnaire are almost the same as previous researches.

Yes, we adapted scales widely used by other studies. The dimensions were kept, only some items were changed to measure readiness for online English learning/ teaching. The focus of this research is to find out how prepared the students/ teachers are for learning/ teaching English online, so we didn’t design a scale from scratch. The creativity lies in the research subjects and the special context. There are few studies focusing on measuring Chinese college students’ and English teachers’ readiness for learning and teaching English online. And the context of COVID-19 was new. However, designing a better scale for measuring readiness for online language learning can be a topic for our future research. 

8. L76: There are differences between pandemic and epidemic. It should be “COVID-19 pandemic”.

The wording has been changed.

9. L85-89: grammatical mistakes. “It is flexible; can be adaptive; allows for enhanced, individualized, and authentic materials; can take advantage of communicative tasks and multilingual communities; can foster and take advantage of autonomous learning and learner corpora”.

The sentence has been rewritten as “First of all, it is flexible, adaptive and allows for enhanced, individualized, and authentic materials. Secondly, it can take advantage of communicative tasks and multilingual communities. Lastly, it can also foster and take advantage of autonomous learning and learner corpora.” 

10. L103 “Many HELs are developing MOOCs and SPOCs for English education online, but are teachers and students ready for the development? Literature in this domain was relatively scare.” Before addressing the research gap, the authors mentioned several literature on online education. However, there are literature related to MOOC for English learning, maybe the authors could add more literature related to online language learning. 

Thank you for your advice and we have rewritten the literature review part to focus specifically on online language learning readiness. 

Methodology：

11. Why there are only student respondents from first- and second-year grade?

Because in China College English is a compulsory course for first- and second-year graders. This has been mentioned in the introduction part. Based on your reminding, we also added related information in “3.3 Data acquisition” to avoid misunderstanding.

12. The specific procedure in which the researchers used the tool, ATLAS.TI 8, for coding needs elaboration.

The coding procedures were added. “Qualitative data obtained from open-ended questions were analyzed using topic and analytical coding (Richards, 2014). Strict procedures were followed to ensure coding reliability. Firstly, the answers to the four questions were uploaded to the qualitative research program ATLAS. ti 8 as separate documents. Secondly, two researchers went through the documents to have a rough idea and started coding independently. The open coding and auto-coding functions were combined, but the auto-coding results were checked to avoid inappropriateness. Categories kept emerging through the process of coding. When this initial stage of independent coding finished, the two researchers compared their codes and categories to negotiate a final version. Lastly, categories were further analyzed.”

13. Please make sure that the sentence “Qualitative data derived from the SRS and TRS were analyzed... as well as establishing...” is grammatically correct.

The long sentence has been simplified as “Quantitative data derived from the SRS and TRS were analyzed using SPSS 20.0 and Microsoft Excel. On the one hand, the overall readiness of students and teachers for online college English learning/teaching during the online migration was calculated. On the other hand, the data were checked to see whether there were significant differences between different demographic groups.”

14. L229: it is not clear that 18 refers to what here?

The sentence has been deleted as the description was improved. 

15. L293-295: there are no statistic supports for the mainstream platform like Tencent Classroom. 

The statistics were added to make the statement convincing: The mainstream ones were Chaoxing learning app and MOOC platform (40.16%), Icourse163 MOOC platform (28.78%), QQ (27.58%), WeChat (26.36%), VooVmeeting (22.16%), Tencent Classroom (21.30%), Dingtalk (18.10%). Teachers generally had their own choices with the tools by negotiating with students but a common phenomenon was a mixture of different tools. Only a minority of teachers (17.65%) managed to stick with only one platform or tool [60].

Findings：

16. L524-526: the authors mentioned that “perceptions for future online English education, some students think online or blended English courses have the advantage of high efficiency and effectiveness”. But in line479-481, “participants also commented on the relatively low efficiency and undesirable effectiveness of the online English classes compared with F2F ones”. Further analysis of what led to this result is needed.

Further analysis was added: “What drew our attention was students’ opposing perceptions towards the efficiency and effectiveness of online English courses. Some students claimed that online English learning had advantages of high efficiency and effectiveness, while others emphasized its low efficiency and ineffectiveness. By looking at the respondents’ scores of readiness and their answers to the open-ended questions, it was found that students who had lower scores of readiness in the self-directed learning, learn control and online communication self-efficacy dimensions generally considered online English courses less efficient and effective. These students usually lack self-discipline and can not regulate their own learning without the teacher’s presence. As a result, they found they were benefiting less from the online English courses. What’s more, students who reported poor internet connection or software and hardware failures also thought online learning less efficient. Contrarily, students who were capable of self-control and used to communicating in English online verbally or in written form viewed high efficiency and effectiveness as advantages of online English learning, because they could have their own pace of learning and have more chances to communicate with others.”

17. In 4.1 the authors mentioned about college English education before, during and after COVID-19. But how to define “after COVID-19”? The overview of college English education before and during COVID-19 was supported by data, but overview of college English education after COVID-19 was not.

The wording was changed (before and during) to avoid misunderstanding because we didn’t carry out follow up interviews to check the real situation in these universities after the pandemic semester was over.

18. L434-436: Why echo with an unpublished paper? 4.3.2 and 4.3.3 are from two open-ended questions. What about the data of interview?

The reason for echoing with that paper was to show teachers concern about student engagement in online classes. The paper was published and was cited in the revised version. 

The data of the interview was analyzed in 4.1 to give an overview of college English education before and during COVID-19. Explanation was added at the beginning in section 4.1: From the interviews, an overall view about college English education in the sampled universities before and during the pandemic was gathered. The information was not directly related to the research questions, but it can help to better understanding the statistical information in the subsequent sections.

19. Concerning the finding in p22 “Teachers generally had their own choices with the tools by negotiating with students but a common phenomenon was a mixture of different tools”. The author could further discuss more about the phenomenon of mix use of different tools. Or if it was from the previous studies, the authors should give clear citations.

Citations were given to statistics quoted and further explanations about the phenomenon of mix use of different tools were also added.

Discussion & implication：

20. L526-529: no discussion for why students’ perception of online courses varies so differently. Some students view online learning has advantages high efficiency and effectiveness. While other student consider online learning has disadvantages: ineffectiveness, low efficiency.

Discussion was added: What drew our attention was students’ opposing perceptions towards the efficiency and effectiveness of online English courses. Some students claimed that online English learning had advantages of high efficiency and effectiveness, while others emphasized its low efficiency and ineffectiveness. By looking at the respondents’ scores of readiness and their answers to the open-ended questions, it was found that students who had lower scores of readiness in the self-directed learning, learn control and online communication self-efficacy dimensions generally considered online English courses less efficient and effective. These students usually lack self-discipline and can not regulate their own learning without the teacher’s presence. As a result, they found they were benefiting less from the online English courses. What’s more, students who reported poor internet connection or software and hardware failures also thought online learning less efficient. Contrarily, students who were capable of self-control and used to communicating in English online verbally or in written form viewed high efficiency and effectiveness as advantages of online English learning, because they could have their own pace of learning and have more chances to communicate with others.

21. Line 578: why the authors said “with the exception of pedagogical readiness...”, while in page 23, especially in item, pedagogical readiness got a low score. It would be better to discuss the low score of item13.

Discussion was added to explain the low score of item 13 (which was now item PR6): Teachers were having a difficulty in engaging students online, and it also was proved by their reported challenges (to be discussed in the following section). In order to make the class go smoothly, teachers would generally require students to turn on the camera and microphone only when answering questions. Without seeing each other, the teacher can not use his or her body language and eye contact to encourage students to participate in activities, and the students can not feel involved, either. Also, the students tend to be distracted without the teacher’s presence, thus participating less in class.

22. L599-602 The expression seems confusing and could be polished.

The sentence was polished: Engagement, interaction, and communication can determine the success of an online course and the learner’s learning performance. 

 

Response to Reviewer #6

Thank you for your positive comments on our manuscripts and valuable suggestions to improve it. We have revised the manuscript based on your comments.

1. The samples in this study were mainly collected from one province, which might be one of the limitations of the study. 

Initially, we believed the data gained from the survey conducted in Wuhan were representative enough to represent the condition of online English education in China for the following reasons. Firstly, according to the statistics released by the Ministry of Education (MOE), Wuhan (the capital city of Hubei Province) has the largest student population and the second biggest number of HEIs in China. Secondly, college English education in China follows similar patterns guided by the College English Curriculum Requirements released by the MOE. 

However, after more careful consideration based on comments from you and other reviewers, we decided to change “in China” into “in Wuhan”. Our reasons are: 1) Though college students in Wuhan come from all over the country, the majority are from Hubei Province because of the college student recruiting policy; 2) There are differences concerning technological development between cities.

2. As for the methodology adopted in this study, although the author used factor analysis to extract several different dimensions (factors) behind items in questionnaire, the author did not make further explanations about those different factors and most of the statistics were conducted to make comparisons between different variables. I think the author should add some explanations to illustrate those different factors before making further comparisons between different variables.

Thank you for pointing out this important limitation of the study. After careful thinking, we decided to delete the part of factor analysis, because the focus of the research is to find out how prepared the students/ teachers are for learning/ teaching English online and the problems they encountered in this special context. The scales used were adapted from previous research. The factors were kept, and only the wording of the items was changed. However, after conducting this research, we found designing a better scale for measuring readiness for online language learning would be a valuable topic for our future research. 

3. The paper is generally written in quality academic style but there are several language issues worthy of revision. I list several issues below (though it is not intended to be exhaustive):

Thank you very much for your careful checks. The mistakes listed were corrected and the language of the manuscript was polished.

P3 HEI ---Higher Education Institutions (HEIs) √

Mooc---Massive Online Open Course (Mooc) √

P5 utilizes---utilize √

“It is flexible; can be adaptive; allows for enhanced, individualized, and authentic materials; can take advantage of communicative tasks and multilingual communities; can foster and take advantage of autonomous learning and learner corpora….”

The author used semicolon after each clause. However, most of the clauses lack subjects. If the author insists on using such semicolon structure, I suggest the author to add subjects to those clauses. Perhaps another choice is to rearrange the structure of the sentence to make it more integrated.

The sentence was restructured: First of all, it is flexible, adaptive and allows for enhanced, individualized, and authentic materials. Secondly, it can take advantage of communicative tasks and multilingual communities. Lastly, it can also foster and take advantage of autonomous learning and learner corpora.

P7 student access ----student’s access √

was----is? √

“...teachers need different skills than those normally employed by tutors trained to teach languages in a F2F classroom…..”

This sounds confusing. I think the author might consider add “and” between “tutors” and “trained”.

The sentence was modified: …teachers need different skills than those who are trained to teach languages in a F2F classroom…

P8 lower level skills as --- lower level skills such as ? √

higher level skills as—higher level skills such as ? √

P13 “The survey lasted for ten days.” I suggest the author to combine this sentence with the previous paragraph instead of leaving it as a separate paragraph.

The previous “3.3 Participant recruitment” and “3.4 Survey dissemination” were rewritten as a single section “3.3 Data acquisition”, so the sentence was deleted. 

P14 “As the SRS and TRS were…” I suggest the author to change “as” into “Since” √

P15 “a combine of” ----a combination of √

P19 “suburb area”---suburban area √

P20 “test were”---test was √

P36 “ ....teachers who have taught online and are planning to teach online or blended English courses.” --- The sentence is ungrammatical.

The sentence was revised: The second group is teachers who have taught or are planning to teach online or blended English courses.

“.... businesses which invest in education technology and want to draw more profits.” --- The sentence is also ungrammatical.

The sentence was revised: The last group is businesses which invest in education technology and want to draw more profits.

4. Page 4: The author mentioned that online English course is different from online courses in other disciplines. Perhaps the author should provide some explanations to illustrate the reason why online English teaching is different and much more difficult to implement.

Thanks for your reminding and explanations were added: Online language learning is different from online learning of other subjects. Unlike other subjects, language is both the medium of instruction and the subject matter of online learning. In the learning process, the learners are supposed to listen, speak, read and write in the language they are learning. Therefore, whether the online learning environment can provide opportunities for the learners to use the language and whether the learners feel free to use it online determines the success of the language course.

5. P6 “Literature in this domain was relatively scarce.” This sounds a little incomplete when the author put it at the end of the paragraph.

The sentence was deleted as the whole literature review section was rewritten to be more focused and provide some theoretical foundation. 

6. Section 4.1 overview of college English education before, during and after COVID-19”

In this part, the author listed the different dimensions derived from the questionnaires. However, it seems that the author didn’t make an explicit explanation about those dimensions listed in table 3.

The data analysis part has been rewritten to make the paper more focused. The previous table 3 was deleted and the dimensions were explained in the literature review part. 

 

Response to Reviewer #7

Thank you very much for your review and valuable suggestions to improve our manuscript. We have revised the manuscript based on your comments.

1. Although the title of the manuscript is “Online College English Education in China against the COVID-19 Pandemic”, the data were collected from one single city in China. It’s hard to say that they can represent the entire country.

Initially, we believed the data gained from the survey conducted in Wuhan were representative enough to represent the condition of online English education in China for the following reasons. Firstly, according to the statistics released by the Ministry of Education (MOE), Wuhan (the capital city of Hubei Province) has the largest student population and the second biggest number of HEIs in China. Secondly, college English education in China follows similar patterns guided by the College English Curriculum Requirements released by the MOE. 

However, after more careful consideration based on comments from you and other reviewers, we decided to change “in China” into “in Wuhan”. Our reasons are: 1) Though college students in Wuhan come from all over the country, the majority are from Hubei Province because of the college student recruiting policy; 2) There are differences concerning technological development between cities.

2. Three research questions were listed in the paper, but their relationship is rather unclear. For example, if participants report a lower level of readiness at certain dimension, it is likely for them to report greater challenges at that dimension. So the first two questions are to a certain extent overlapping. I suggest the authors reconsider the research questions to avoid overlapping, and if there is any change in the research questions, the entire paper needs to be updated accordingly. It is also necessary to explain clearly why the third research question needs to be explored.

Thank you for giving us insights about the research questions. There is indeed some overlapping in the first two questions, but we intended to have a rough idea about students’ and teachers’ readiness levels through the scales, and to find out the specific problems they met in this special context. If the problems are specific to the context of COVID-19, students and teachers don’t need to worry. If not, students, teachers and other personnel related should work together to solve the problems in order to facilitate online English learning/ teaching. Also, by analyzing data, we also found inconsistencies between the reported level of readiness and challenges. For the third question, we thought it necessary because the readiness levels and the challenges didn’t necessarily determine the participants willingness to continue learning English online. 

The research question part has been revised to make it clearer for readers, but we are sorry that we could not change the research questions. If we change them, it means we need to do the whole research once again, but the context has already changed. We will be more careful about research questions for our future research. Thanks a lot for your suggestions.

3. Although the authors indicate that “All relevant data are within the manuscript and its Supporting Information files” in the Data Availability Statement, I cannot find the data in the manuscript, and it seems no relevant supporting files have been provided. PLOS ONE requires the authors to share their data publicly, but right now I can only find the statistical results which are not the same as the data. Therefore, I strongly recommend the authors to upload their source data.

Our source data files have been uploaded as required.

4. The abstract fails to summarize the key findings from the research. For instance, what are the challenges encountered by the students and teachers?

Thank you for your advice, and the abstract has been rewritten: 

A survey of 2310 non-English-major college students and 149 English teachers from three types of twelve higher education institutions in Wuhan was conducted to evaluate their readiness for online English education during the COVID-19 pandemic, to figure out challenges they encountered and to draw implications for future online college English education. Quantitative statistics gathered using two readiness scales adapted from previous studies showed that both cohorts were slightly below the ready level for the unexpected online transition of college English education. The overall level of readiness for students was 3.68 out of a score of 5, and that for teachers was 3.70. Individual differences were explored and reported. An analysis of qualitative results summarized six categories of challenges encountered by the students, i.e. technical challenges, challenges concerning learning process, learning environment, self-control, efficiency and effectiveness, and health concern. Though the students reported the highest level of readiness in technology access, they were most troubled by technical problems during online study. For teachers, among three types of challenges, they were most frustrated by pedagogical ones, especially students’ disengagement in online class. Qualitative data also brought insights for online college English education development. Institutions should take the initiative and continue promoting the development of online college English education, because the majority of respondents reported their willingness and intention to continue learning/teaching English in online or blended courses in the post-pandemic period. Technical barriers should be removed, readiness evaluation and instructor training are also necessary. Language teachers are suggested to pay special attention to students’ engagement and communication in online courses.

5. P2 line22-23: “some challenges coherent with their low scores in certain dimensions from the readiness scales”

Since the authors did not mention what the challenges entail, there is no way for readers to understand the meaning of the sentence.

The abstract was rewritten, and this sentence was deleted.

6. P2 line 24-27: “Qualitative data also showed prospects of growing development of online college English education as the majority of respondents reported their willingness and intention to continue learning/teaching English in online or blended courses in the post-pandemic period.”

This conclusion is not convincing. It is oversimplified to conclude that online college English education has potential to grow only on the basis of the fact that the respondents reported their willingness and intention.

The sentence has been improved: Qualitative data also brought insights for online college English education development. Institutions should take the initiative and continue promoting the development of online college English education, because the majority of respondents reported their willingness and intention to continue learning/teaching English in online or blended courses in the post-pandemic period. Technical barriers should be removed, readiness evaluation and instructor training are also necessary. Language teachers are suggested to pay special attention to students’ engagement and communication in online courses.

7. If the participants were anonymously engaged in the survey, and they were not compensated in any way, how can the authors ensure they were sufficiently motivated to provide genuine answers to the questions? In designing the scales, did the authors include any item to detect lies? Have the quantitative data been filtered in any way before the analysis?

The participants were not compensated because we hadn’t got enough fund. We tried to ensure the genuineness of answers from participants through two measures. Firstly, the invitation was sent by English teachers to their students. Secondly, when sending the invitation, the teachers also sent the following information: the survey was anonymous and voluntary; it’s OK if you don’t want to participate in the survey, but if you are willing to help, please give honest answers to the questions. 

It’s shameful that we forgot to include an item to detect lies. We will be conscientious in future surveys. 

The qualitative data were filtered before analysis. We have added the process in the data acquisition part: Among 2351 students who completed the student questionnaire, 15 were from universities from which no interview was conducted, and 26 completed the questionnaire in less than 60 seconds (the researchers tried to finish the questionnaire as fast as they can and determined the minimum completion time acceptable should be 60 seconds). These 41 answers were deleted before analysis. For the teacher questionnaire, 151 teachers completed it, and 2 of them in less than 45 seconds (with the same determining method mentioned above), which was deleted as invalid for further analysis. Therefore, a sample of 2310 first/second-year students and 149 college English teachers from 12 HEIs (3 directly under MOE, 6 under Hubei Provincial Department of Education and 3 non-governmental, diverse in disciplinary areas, student enrollment numbers and include both research and teaching ones) was generated.

8. P6, line 105: “Literature in this domain was relatively scarce.”

Does this indicate there is no literature on this topic at all, or there are only a small number of them? Please make it clear.

This sentence was deleted and we have reviewed more studies related to online language learning readiness in the literature review part in order to make a statement of the research gap.

9. P7 line 126: “few studies have focused on student readiness specifically for online language learning.”

But there are indeed a number of studies on student readiness for online language learning (e.g., Tylor &David 2013; Luu & Lian, 2020; Mehran, et al., 2017). I suggest the authors review the prior studies in a more comprehensive way. The relevant studies should be cited. More importantly, the authors should specify what new contribution this study has made to this field, given all the previous studies that have already been conducted. In reviewing the prior studies concerning teacher readiness, the authors also need to point out the inadequacy of these studies had or the issues they failed to address, which make it necessary for the authors to carry out their own study.

Thanks a lot for your recommendation. We have reviewed these studies and some more on online language learning readiness. The two sections “Student readiness for online language learning” and “Teacher readiness for online language teaching” were rewritten according to your suggestions. 

10. The following methodological concerns need to be addressed in the study:

What is the purpose of combining qualitative and quantitative approaches in this study?

What methodological framework did the study follow?

The quantitative approach is to obtain the overall level of readiness among students and teachers, while the qualitative method is to probe into the problems of the low scores of readiness. The problems reported by the participants help to better explain their self-determined readiness.

Information about theoretical foundation (such as TAM, definitions of the components mentioned in the readiness scales) was added in section 2 Literature review.

11. A semi-structured interview seems to be conducted in this study, but the findings from the interview do not seem to be relevant to the research questions the authors intended to address. The authors need to reconsider the purpose of having a semi-structured interview? If it is necessary to have the interview, the relevant details need to be reported: How was it designed? What procedures were followed? Did the participants answer any question? How were the answers analyzed?

The semi-structured interview was not directly related to the research questions. It is designed to provide extra information to better explain the results from the questionnaires. Relevant details were added in Section 3.2 Instrument design and 3.3 Data acquisition. 

12. The data analysis section is not helpful for readers to understand how the data analysis was made. The authors only introduced the tools for statistical analysis, which is far from enough. They should also report the statistical approaches used in the analysis, how data trimming was performed, the procedures of data analysis, how the interview was transcribed, how they ensured the accuracy of transcription, etc. The statements about research instrument should be moved to the Instrument section.

The instrument section, data acquisition section and data analysis section were rewritten according to your suggestions. 

13. The results section was not written in a logical way. I suggest the authors reorganize it to respond to the three research questions. Section 4.1 does not seem to be relevant to any research question. Also the relation between 4.2.1 and 4.2.2 is not clear. 

Section 4.1 does not respond to any research question. It was written to summarize the information from the interviews, and some information can provide extra explanation to the statistical findings.

The previous 4.2.1 focus on the overall level of readiness of students, and 4.2.2 compares the differences between different groups of students, for example, students from urban and rural areas.

14. The statistical results were not reported in a standard way. For all the results from t test or ANOVA, degree of freedom and effect size should be reported. For any insignificant results (e.g., line440-441), the authors should report the statistical results with the exact p values. All statistical symbols (sample statistics) that are not Greek letters should be italicized.

The Findings part was greatly improved according to your advice. The format was also improved to be in line with common practice. Thank you for being nice to pointing out every mistake that we had neglected. 

15. The quantitative study shows that students were most ready in terms of technology access, which seems to contradict the finding from the qualitative study which indicates that the greatest challenge for students was technical. Does this divergence reflect an inadequacy in the design of the student readiness questionnaire? More explanation is needed.

Explanation on the discrepancy was added in section 5.1: The students participated in this survey belong to the so-called generation of digital natives. In accordance with this, they rated themselves ready in the technology access dimension. The majority of them own or have access to computers, smartphones and the internet. Nevertheless, the stability of internet connection differs from area to area, especially when all the students were learning online. That could explain the discrepancy between students self-determined readiness level in technology access and the frequently reported problems in internet connection.

16. P27 line 489-491: Despite the teachers’ higher scores of pedagogical readiness in the TRS, their main difficulties during the online semester pertained to pedagogical issues…

Please account for the discrepancy.

This sentence was deleted in the improved manuscript, and the discrepancy was explained in 5.1: What need to be discussed were teachers’ relatively high level of pedagogical readiness and the frequently reported pedagogical challenges. Teachers claimed to be positive towards online English teaching and were willing to learn new technology and skills. They also thought themselves capable of designing online English classes. However, during the pandemic semester, their primary concern was the outcome of teaching. In line with their low readiness score in students’ engagement, the biggest challenge they met was how to effectively engage students in online classes. The problem was not only perceived by the teachers, but also reported by the students as bad communication and poor interaction….

17. In the Results section, the authors have reported the results concerning the individual differences in students or teachers, but very few of them were summarized and discussed in the Discussion section. The discussion is rather inadequate.

The Discussion section was rewritten to summarize and explain all the findings. Thank you for reminding us.

18. The language of the manuscript is below the standard for publication. It contains too many grammatical and other errors, and some sentences are very difficult to understand. It needs to be proofread thoroughly and carefully by a native speaker of English. Below I’ve listed some of the problems, but such problems are almost everywhere:

Thank for very much for pointing out the language errors. The manuscript was proofread and the language was polished. Language problems, including those listed, were corrected. 

P5 86-89: “It is flexible; can be adaptive; allows for enhanced, individualized, and authentic materials; can take advantage of communicative tasks and multilingual communities; can foster and take advantage of autonomous learning and learner corpora”

The sentences are fragmented.

The sentence was rewritten as: “First of all, it is flexible, adaptive and allows for enhanced, individualized, and authentic materials. Secondly, it can take advantage of communicative tasks and multilingual communities. Lastly, it can also foster and take advantage of autonomous learning and learner corpora.”

P6 line98: in infancy level > at its infancy √

P6 line113: either fully or hybrid > either fully or hybridly √

P7 line129: Concerning instructors, teaching online learning requires a reconstruction of their…

What does “teaching online learning” mean?

The word “learning” was deleted.

Line550: understanding how was the situation>understanding the situation √

Line 613: there are 3 types interaction

> there are 3 types of interaction

 

Response to Reviewer #8

Thank you very much for your careful review and valuable suggestions. We have revised the manuscript based on your comments.

1. pp.10-11 “The two questionnaires were initially piloted to check clarity of the language used and to ensure the reliability and validity of the two scales in the local context. Improvements were made in light of the comments from pilot respondents and two experts in research methodology. Both scales were statistically reliable and valid with the pilot tests.”

-- Could you please describe in more detail what “improvements” or modifications you did on the questionnaires? In addition, what are the results of the statistical analyses for the evaluation of the reliability and validity of the questionnaires in the pilot tests?

This part was rewritten as: The two questionnaires were initially piloted to check clarity of the language used and to ensure the reliability of the two scales in the local context. Both scales were statistically reliable, with Cronbach’s Alpha coefficients being 0.961 and 0.859 respectively. Improvements were made in light of the comments from pilot respondents and two colleagues with expertise in questionnaire design. Improvements included removing one item which was overlapping with another one from the TRS and clarifying language ambiguities. For instance, one item “当我的电脑软硬件出现技术问题时，有人和/或资源给我提供帮助。” was modified as “在线教学过程中出现技术问题时，有人（同事，家人，平台技术支持）和/或资源（手册，视频）给我提供帮助。” 

2. p.11 “a semi-structured interview was conducted between the researchers and the personnel in charge of college English education from the sampled universities or colleges to get an overall view of the situation before, during and after the pandemic semester for the sake of better understanding of the statistical results of the survey.”

-- Could you describe in detail how the interview was administered? In particular, how did you choose the participants of the interview? What was the procedure of the interview?

The participants of the interview were invited by the director of our department. These participants are in charge of the college English education in their universities, so they know the details of the online English teaching during the pandemic semester. Thanks to your advice, the procedure of the interview was added in the manuscript. Relevant information was underlined in the following:

In order to involve universities at all levels, the director of the School of Foreign Languages from the researchers’ university sent invitations to her counterparts from 19 institutions (4 directly under MOE, 10 under Hubei Provincial Department of Education and 5 non-governmental) purposively to recruit teacher and student respondents who teach or learn college English courses. All of them replied and accepted the invitation to help without promising a satisfactory result because participation was voluntary and responses were anonymous.

The questionnaires were disseminated online from July 24th to August 2nd, 2020 after the semester terminated in all universities. Both were posted on one of the most widely-used online survey platforms in China powered by https://www.wjx.cn/ and only those invited by their college English teachers got the access to participate. The survey was set to allow only one submission for the sake of data integrity. 

The interviews were conducted during the same time period by the researchers with the personnel in charge of college English education from the sampled universities or colleges on a one-to-one basis through email or online chatting tools. Written answers were copied directly. Oral ones were transcribed automatically by chatting tools first and then checked by one of the researchers before further analysis.

3. p.14 “Cronbach’s alpha coefficients were calculated to ensure internal consistency and confirmatory factor analysis was carried out to provide evidence for convergent validity.”

-- The readers may also be curious of other results of the CFA than those you reported in Table 3. Particularly, what is the overall result/significance of the model?

The corresponding part was rewritten to exclude details about CFA. We used scales validated in previous studies with pre-determined dimensions, and the focus of the study was not to validate the scales, so we didn’t carry out EFA with the pilot sample. We intended to show details of CFA with the main sample, and the KMO values indicated that the samples were adequate for CFA, but the final results were problematic. That’s why we only included information of AVE and CR in previous Table 3. However, we still wanted to show the results of the survey, so we changed our way of analysis for a little bit and exclude details about CFA. We are sorry for this limitation, and we will be more conscientious with future studies.

 

Response to Reviewer #9

Thank you very much for your careful review and valuable suggestions. We have revised the manuscript based on your comments.

Introduction and theoretical framework:

1. On page 4. The author mentioned that “…lessons can be and should be drawn for future development of online college English education”. It is necessary for the authors to clarify why and how the examination of emergency remote learning that occurred during Covid-19 can hold implications for that in the non-pandemic periods. Remote learning during a pandemic has characteristics that are not present during non-pandemic periods.

Thank you for pointing out the problem. The part has been rewritten: It was different from well-prepared and planned online teaching. Generally speaking, for college English courses in China, online learning is currently acting as a complementary means to classroom teaching. Learning platforms coming along with the textbooks and self-developed MOOCs or Small Private Online Courses (SPOCs) are the mainstream tools for English teachers to implement online or blended teaching. Problems may occur, but at a low frequency and do not cause too many anxieties because there is classroom teaching. However, during the pandemic, online English learning was the only compulsory means rather than a complementary one. Problems occurred frequently, especially at the beginning, and caused anxieties on both the teaching and learning sides. Some problems might be specific to the pandemic context, others might be common ones even in non-pandemic period. Therefore, now that the semester has terminated smoothly and successfully, lessons can be and should be drawn for future development of online college English education. This research aims to draw implications for the development of online college English education through measuring the readiness levels of the students and teachers for the online transition and probing into the problems they met in this particular context.

2. In the section titled theoretical framework, it seems that the authors did not elucidate clearly or even did not elucidate any theories behind this study. The authors need to consider introducing a theory (theories) supporting this study.

The section was renamed as Literature review, and we have added relevant information (such as TAM and definitions of major components of a readiness scale” as the theoretical foundation.

3. The empirical studies reviewed are not those conducted during the period of Covid-19. The authors need to consider focusing the literature review on or most of the reviewed should be from the pandemic period.

We have added a few studies conducted during the pandemic in section 2.4 Online language teaching during COVID-19 pandemic.

4. On page 6. The author mentioned “…but are teachers and students ready for this development (developing MOOCs and SPOCs)?” I think there are some studies exploring student and teacher readiness for MOOCs and SPOCs. Also, there is a difference between readiness for MOOCs and that for emergency remote learning. Authors should focus on reviewing studies that examined remote learning during emergency situations.

We have rewritten the whole literature review part to make it more focus and provide some theoretical foundation. Studies on online language learning during the pandemic were also reviewed in 2.4 Online language teaching during COVID-19 pandemic.

Methodology:

1. Authors need to clarify how the questionnaires were translated into Chinese?

Clarification was added in the instrument design part: Translation of the questionnaires were done by one of the researchers who has a master’s degree in translation, and double checked by a colleague with a master’s degree in translation.

2. Authors need to explain how the data were filtered and how invalid data were removed.

Clearer explanation on how data were filtered was added: Throughout the survey, personnel in charge of college English education from 16 universities participated in the interview. Among them, 4 were excluded because researchers received no teacher response or less than 10 student responses from these universities. Among the 2351 students who completed the student questionnaire, 15 were from universities from which no interview was conducted, and 26 completed the questionnaire in less than 60 seconds (the researchers tried to finish the questionnaire as fast as they can and determined the minimum completion time acceptable should be 60 seconds). These 41 answers were deleted before analysis. For the teacher questionnaire, 151 teachers completed it, and 2 of them in less than 45 seconds (with the same determining method mentioned above), which was deleted as invalid for further analysis. Therefore, a sample of 2310 first/second-year students and 149 college English teachers and 2310 first/second-year students from 12 HEIs (3 directly under MOE, 6 under Hubei Provincial Department of Education and 3 non-governmental, diverse in disciplinary areas, student enrollment numbers and include both research and teaching ones) was generated.

3. What is the purpose of conducting confirmatory factor analysis on both pilot and final sample? Typically, we use the pilot sample for exploratory factor analysis and the main sample for confirmatory factor analysis.

The corresponding part was rewritten to exclude details about CFA. We used scales validated in previous studies with pre-determined dimensions, and the focus of the study was not to validate the scales, so we didn’t carry out EFA with the pilot sample. We intended to show details of CFA with the main sample, and the KMO values indicated that the samples were adequate for CFA, but the final results were problematic. That’s why we only included information of AVE and CR in Table 3. However, we still wanted to show the results of the survey, so we changed our way of analysis for a little bit and exclude details about CFA. We are sorry for this limitation, and we will be more conscientious with future studies.

Findings:

1. Section 4.1 could be written more concisely. And it can be considered in the methodology section as research background.

It’s a nice suggestion to put 4.1 in the methodology section as research background. We tried, but failed. 4.1 was a summary of the interviews, it actually belongs to results. We tried to delete the interviews, but we were afraid that readers would doubt how we obtained the information. Therefore, we kept it unchanged, but deleted some information.

2. The results would be more informative if demographic characteristics were in a regression model as predictor variables to assess their predictive power on each indicator of readiness. It seems that prior experiences with online education should be an important factor influencing readiness. Did the authors consider this factor?

We appreciated your advice about the regression model, but we prefer not to further complicate the research in this paper any more. And thank you for your reminding about prior experiences as a factor. Upon your suggestion we added relevant information in the discussion part: The study failed to consider previous experience with online education as a variable, but students coming from different types of institutions were found to have different levels of readiness. The reason might be the fact that the three sampled non-governmental universities relied fully on traditional teaching for college English education while the others had already implemented blended teaching or online learning prior to the pandemic.

3. The authors examined whether there were significant differences in overall readiness by grade and gender, but what about each indicator of readiness?

We have checked, and there was no statistically significant difference by grade and gender for each indicator of readiness. We didn’t report it because we don’t want to make it too complicated. If there were differences, we would have reported. 

4. The authors examined differences in readiness for online education across disciplines. However, as I understand it, the authors wanted to examine the readiness of students for online English education. So why is it important to examine readiness for English online classes across disciplinary backgrounds?

In China, all college students, except those majoring in English, have to study college English. According to the researchers teaching experience, students with different disciplinary backgrounds have different levels of English language efficacy and different attitudes towards English learning. That’s why individual differences were examined across disciplinary backgrounds.

5. The authors examined both student and teacher readiness. And what is the connection between them? Why is it important to discuss both the teachers’ and the students’ readiness in one article if the connection between was not explicitly examined?

Thank you for your doubt. To be honest, we regret for the decision to discuss both, which made the whole research much more complicated than we thought. However, we had come so far, so we didn’t want to quit. Our justification was that the teacher and the students acting together determine the success of an online course, so readiness levels from both cohorts were important. It is a possible topic to explore the connection between the teachers’ and the students’ readiness for future research.

6. The authors used interviews to examine the challenges students faced. I wondered why the purpose of the interviews was not to examine the factors that influence success and failure in readiness for Covid-19 emergency remote learning. This would make the article more focused.

We thought that success and failure would be better determined by the learning outcomes, such as a test, so we made it more objective to ask respondents to report the challenges. However, we are going to discuss about it for possible future studies.

7. The research question addressed in section 4.3.3 was “Are you willing to learn/teach English …in the future?” How would authors define the future? The future with a pandemic or emergency situations? Or a future without any emergency situations?

This is our fault. We didn’t specify “the future”, and we didn’t realize the problem until the we’ve got the final results, because a number of respondents mentioned willingness to have online classes in emergencies but still prefer traditional classes in normal situations. For the pilot tests, we only got a sample of 37 teachers and 104 students, so the results didn’t show the problem. Thank you for pointing out this problem, and we have added this information as a limitation of the study. We are sure to be more cautious for future instrument design.

Discussion

1. The discussion needs to be deeply integrated with the relevant literature conducted during Covid-19.

Thank you for the suggestion, and we have integrated a few literatures conducted during the pandemic. 

2. The pedagogical implications should be stated more specifically and concretely.

The pedagogical implications were stated more specifically in the new section 5.2 Implications for educational settings.

 

Response to Reviewer #10

Thank you very much for your careful review and valuable suggestions. We have revised the manuscript based on your comments.

1.The authors failed to evaluate the context of the nation-wide online teaching in a justifiable way. They claimed that “This online teaching in the face of COVID-19, or exactly, online triage (Gacs et al. 2020), was carried out without need analysis or readiness evaluation from both learning and teaching sides and was different from well-prepared and planned online teaching.” But actually it was not the case: before the epidemic, online teaching had already serve as a positive supplement to offline teaching, and it is available and accessible for students in most part of China, both in rural and urban areas; there are many university websites and apps which offer free online courses, which quite a number of students make use of regularly in their spare time; great efforts have been made by education authorities, especially the Ministry of Education, to analysis and evaluate the situation, to invest considerably in providing online teaching resources, and to mobilize the education-related IT companies to give a helping hand; schools and teachers had tried their best to make preparations before the launch of the online courses. In short, this nationwide online teaching experience was not unprepared and unplanned at all. So, the authors should adjust their wording throughout the whole paper, in order not to make the basis of this research seemingly groundless.

Thank you for pointing out the problem. The advancement in online education in China, and the efforts by MOE and schools were mentioned in the introduction part, but a careful check throughout the paper still showed a lot of inappropriate diction, as you said. Inappropriate wording has been adjusted to make the paper more objective.

2. Get the paper checked by expert speakers. The sentences are extremely long, which makes reading the text challenging. And occasionally, there are some inappropriate wording or even grammatical mistakes. All these may reduce the readability of this paper to some extent.

The language of the whole paper was polished.

3.The teacher sample is relatively small, which makes the analysis and results less convincing.

Yes, the teacher sample was small because participation was voluntary and we wanted to include participants from as many institutions as possible. We asked the directors of the College English Departments to send the invitation for only one time, instead of urging the teachers to participate. For future research, we will give compensation to involve more participants. 

4.The analysis would be more reasonable and sound, if the subjects of teachers and students were divided into two groups, rural and urban.

The subjects of teachers were not divided into rural and urban groups, because all these institutions are located in cities. In Table 4, we analyzed the difference of readiness levels between students from rural and urban areas. 

5. It would be better to make the review of literature more pertinent to online English teaching. In the present form, this paper focuses too generally on online teaching as a whole. Thus it is advisable for the authors to make a very brief review of general online teaching but to concentrate more on English teaching, just as the authors claim in the paper that online teaching is carried out differently to cater for the need of different subjects or disciplines.

Thank you for your advice. The literature review section was rewritten to make it more pertinent to online language education, especially readiness for online language learning and teaching.

---

## [Decision Letter · Decision Letter 1]

10 Sep 2021

**Comments to the Author**

1. If the authors have adequately addressed your comments raised in a previous round of review and you feel that this manuscript is now acceptable for publication, you may indicate that here to bypass the “Comments to the Author” section, enter your conflict of interest statement in the “Confidential to Editor” section, and submit your "Accept" recommendation.

Reviewer #1: (No Response)

Reviewer #5: (No Response)

Reviewer #6: All comments have been addressed

Reviewer #8: All comments have been addressed

2. Is the manuscript technically sound, and do the data support the conclusions?

Reviewer #1: Yes

Reviewer #5: Partly

Reviewer #6: Yes

Reviewer #8: Yes

3. Has the statistical analysis been performed appropriately and rigorously? 

Reviewer #1: Yes

Reviewer #5: N/A

Reviewer #6: Yes

Reviewer #8: Yes

4. Have the authors made all data underlying the findings in their manuscript fully available?

Reviewer #1: Yes

Reviewer #5: Yes

Reviewer #6: Yes

Reviewer #8: Yes

5. Is the manuscript presented in an intelligible fashion and written in standard English?

Reviewer #1: Yes

Reviewer #5: Yes

Reviewer #6: Yes

Reviewer #8: Yes

6. Review Comments to the Author

Reviewer #1: Thank you for allowing me to review your manuscript again. This time the authors have exerted many efforts to enhance their manuscript. Indeed, the manuscript in this version is better than that in the previous version. Since the manuscript has presented major revisions in terms of the overall arrangement, the authors can also make further improvements to make the manuscript clearer and more persuasive.

First, the authors can further emphasize the central part of “readiness”. Since this concept is the keyword, the authors can highlight the significance of “readiness” in online EFL.

Besides, the authors can further stress their contribution to technological development and pedagogical practices to demonstrate the practicality of this manuscript. The potential readers can also grasp the persuasive information concerned with the target field.

At last, to guarantee a more explicit manuscript structure, the authors can answer the research questions or hypotheses in the Conclusion section. Thus, this manuscript can tell the readers what have been done to investigate particular questions.

Based on the previous evaluation, the authors can provide a minor revision to make a more excellent and persuasive manuscript.

Reviewer #5: The author did improve the quality and the readability, however, I'm still not confident to recommend this manuscript to be published in PLOS ONE, the most imporant reason is that it's no longer the right time to publish articles related to people's readiness and challenges during the COVID-19, since it's already the post-pandemic era now.

Reviewer #6: In the abstract, the sentence "Technical barriers should be removed, readiness evaluation and instructor training are also necessary." seems ungrammatical. It needs to be further polished. In addition, the abstract starts with the method part, which is OK. But I think it is better to give some background introduction at the very beginning of the abstract to illustrate the research domain.

Reviewer #8: PONE-D-21-14306_R1

The authors have well addressed the concerns of the reviewers, and I very much appreciate what the authors have done in the revision.

With that said, I would suggest the authors closely read the manuscript again, although the language has very much improved compared with that of the last version. One example lies in Line 55, p.4, where "at a short notice" seemingly should be "at short notice" (see https://www.collinsdictionary.com/dictionary/english/at-short-notice and https://www.ldoceonline.com/dictionary/at-short-notice).

7. PLOS authors have the option to publish the peer review history of their article (what does this mean?). If published, this will include your full peer review and any attached files.

Reviewer #1: **Yes: **Yu Zhonggen

Reviewer #5: No

Reviewer #6: No

Reviewer #8: No

---

## [Author Response · Author response to Decision Letter 1]

13 Sep 2021

Response to Reviewer #1

Thank you for allowing me to review your manuscript again. This time the authors have exerted many efforts to enhance their manuscript. Indeed, the manuscript in this version is better than that in the previous version. Since the manuscript has presented major revisions in terms of the overall arrangement, the authors can also make further improvements to make the manuscript clearer and more persuasive.

Thank you very much for reviewing our manuscript again and recognition of our efforts in improving the manuscript. The improvements couldn’t have been made without your suggestions and those from other reviewers. We are grateful for your further suggestions and the problems are addressed as follows.

First, the authors can further emphasize the central part of “readiness”. Since this concept is the keyword, the authors can highlight the significance of “readiness” in online EFL.

Relevant studies were summarized to highlight the significance of “readiness” in online EFL in Section 2.2: 

Several studies concluded the significance of e-readiness in online learning from different perspectives. Moftakhari [32] claimed the success of online learning entirely relied on learners’ and teachers’ readiness levels, which seems to be absolute. Piskurich [33] believed low readiness level was the main reason for failure in online learning. Students’ e-learning readiness was proved statistically as a significant predictor of their satisfaction for online instruction [34-36]. Therefore, assessing student readiness for online learning is highly relevant prior to delivering a course online either fully or hybridly and promoting student readiness is essential for successful online learning experiences [37].

Besides, the authors can further stress their contribution to technological development and pedagogical practices to demonstrate the practicality of this manuscript. The potential readers can also grasp the persuasive information concerned with the target field.

Implications for teaching staff, institutions and IT companies were listed in Section 5.2. Upon your advice, we checked the implications to see whether they corresponded with the findings and made some improvements.

At last, to guarantee a more explicit manuscript structure, the authors can answer the research questions or hypotheses in the Conclusion section. Thus, this manuscript can tell the readers what have been done to investigate particular questions.

The conclusion part was rewritten to answer the research questions briefly in order to achieve a more explicit manuscript. Detailed explanations were included in Sections 4.2-4.4 and Section 5.1, so the conclusion part was kept concise. 

 

Response to Reviewer #5

The author did improve the quality and the readability, however, I'm still not confident to recommend this manuscript to be published in PLOS ONE, the most important reason is that it's no longer the right time to publish articles related to people's readiness and challenges during the COVID-19, since it's already the post-pandemic era now.

Thank you for your favorable comments on our revised version. We were aware of the timeliness of the submission, but it took us much longer time to analyze the data and write the draft than expected. Unfortunately, the pandemic still persists in many parts of the world, though it’s already the post-pandemic era and people have been trying to live with the virus. Here in China, classes were moved online occasionally in several cities when local cases were confirmed, and many Chinese students who were enrolled in international programs were having remote classes due to restrictions on international travel. In addition, though the research was conducted during the pandemic and the analysis focused on people’s readiness and challenges during that period, the results can provide some implications for online English teaching in general as indicated in the manuscript. Therefore, we sincerely hope to get an opportunity to publish the research. Moreover, thanks for your inspiration, we may explore more issues in online language teaching in the post-pandemic era. 

 

Response to Reviewer #6

1. In the abstract, the sentence "Technical barriers should be removed, readiness evaluation and instructor training are also necessary." seems ungrammatical. It needs to be further polished. 

Thank you for carefully reviewing our manuscript again. The sentence was ambiguous and it was rewritten as: They are supposed to remove technical barriers for teachers and students, and assess the readiness levels of both cohorts before launching English courses online. Institutions should also arrange proper training for instructors involved, especially about pedagogical issues.

The language of the manuscript was polished again upon suggestions from you and another reviewer.

2. In addition, the abstract starts with the method part, which is OK. But I think it is better to give some background introduction at the very beginning of the abstract to illustrate the research domain.

Thanks for your advice and we take it. The following sentence was added at the beginning of the abstract: 

Online education, including college English education, has been developing rapidly in the recent decade in China. Such aspects as e-readiness, benefits and challenges of online education were well-researched under normal situations, but fully online language teaching on a large-scale in emergencies may tell a different story.

 

Response to Reviewer #8

The authors have well addressed the concerns of the reviewers, and I very much appreciate what the authors have done in the revision.

With that said, I would suggest the authors closely read the manuscript again, although the language has very much improved compared with that of the last version. One example lies in Line 55, p.4, where "at a short notice" seemingly should be "at short notice" (see https://www.collinsdictionary.com/dictionary/english/at-short-notice and https://www.ldoceonline.com/dictionary/at-short-notice).

Thank you for reviewing our manuscript again and giving us positive comments on our last revision. We are so grateful because the manuscript couldn’t have been improved without constructive suggestions from you and other reviewers. 

We feel ashamed of our silly mistakes (like the one you pointed out), and the language was polished again. Hopefully it has been improved. We are truly grateful for and inspired by your meticulousness.  

Response to Journal Requirements

The reference list was checked again and no retracted paper was cited. However, when revising the manuscript, two new references were added and the order of several references was changed. The reference list of this revised version was complete and correct.

---

## [Editor Report · Decision Letter 2]

20 Sep 2021

Online College English Education in Wuhan against the COVID-19 Pandemic: Student and Teacher Readiness, Challenges and Implications

PONE-D-21-14306R2

Dear Dr. Li,

We’re pleased to inform you that your manuscript has been judged scientifically suitable for publication and will be formally accepted for publication once it meets all outstanding technical requirements.

Kind regards,

Di Zou

Academic Editor

PLOS ONE
---

## [Editor Report · Acceptance letter]

24 Sep 2021

PONE-D-21-14306R2 

Online College English Education in Wuhan against the COVID-19 Pandemic: Student and Teacher Readiness, Challenges and Implications 

Dear Dr. Jin:

I'm pleased to inform you that your manuscript has been deemed suitable for publication in PLOS ONE. Congratulations! Your manuscript is now with our production department. 

Kind regards, 

on behalf of

Dr. Di Zou 

Academic Editor

PLOS ONE